# 334-year coral record of surface temperature and salinity variability in the greater Agulhas Current region

Jens Zinke[*,1,2,3,4], Takaaki K. Watanabe[5,6], Siren Rühs[7], Miriam Pfeiffer[5], Stefan Grab[4], Dieter Garbe-Schönberg[5], Arne Biastoch[7]

[1]School of Geography, Geology and the Environment, University of Leicester, Leicester, LE1 7RH, United Kingdom
[2]Molecular and Life Sciences, Curtin University, Perth, WA 6102, Australia
[3]Australian Institute of Marine Science, Townville, QLD 4810, Australia
[4]School of Geography, Archaeology and Environmental Studies, University of Witwatersrand, Witwatersrand, South Africa
[5] Institute for Geosciences, University of Kiel, Kiel, 24118, Germany
[6]KIKAI Institute for Coral Reef Sciences, Kikai Town, Kagoshima 891-6151, Japan.
[7] GEOMAR Helmholtz Centre for Ocean Research Kiel, Kiel, 24105, Germany

*Correspondence to*: Jens Zinke (jz262@leicester.ac.uk)

**Abstract.**

The Agulhas Current (AC) off the southern tip of Africa is one of the strongest western boundary currents and a crucial chokepoint of inter-ocean heat and salt exchange between the Indian and the South Atlantic Ocean. However, large uncertainties remain concerning the sea surface temperature (SST) and salinity (SSS) variability in the AC region and their driving mechanisms over longer time scales, due to short observational datasets and the highly dynamic nature of the region. Here, we present an annual coral skeletal Sr/Ca composite record paired with an established composite oxygen isotope record from Ifaty and Tulear reefs in southwestern Madagascar to obtain a 334 year-long (1661-1995) reconstruction of $\delta^{18}O_{seawater}$ changes related to surface salinity variability in the wider Agulhas Current region. Our new annual $\delta^{18}O_{seawater}$ composite record from Ifaty traces surface salinity of the southern Mozambique Channel and AC core region from the SODA reanalysis between 1958 and 1995. $\delta^{18}O_{seawater}$ appears mainly driven by large-scale wind forcing in the southern Indian Ocean on interannual to decadal time scales. The $\delta^{18}O_{seawater}$ and SST at Ifaty show characteristic interannual variability of between 2 to 4 years and interdecadal 8 to 16 years variability, coherent with El Nino-Southern Oscillation (ENSO) records. Lagged correlations with the Multivariate ENSO index reveals a 1-2 year lag of $\delta^{18}O_{seawater}$ and salinity at Ifaty and the AC region, suggesting that propagation of anomalies by ocean Rossby waves may contribute to salinity changes in the wider southwestern Indian Ocean. The $\delta^{18}O_{seawater}$ and SST reconstructions at Ifaty reveal the highest interannual variability during the Little Ice Age, especially around 1700 CE, which is in agreement with other Indo-Pacific coral studies. Our study demonstrates the huge potential to

unlock past interannual and decadal changes in surface ocean hydrology and ocean transport dynamics from coral $\delta^{18}O_{seawater}$ beyond the short instrumental record.

## 1 Introduction

The greater Agulhas Current (AC) system off the southern tip of Africa is a crucial chokepoint of the global thermohaline circulation through inter-ocean heat and salt exchange between the Indian and the South Atlantic Ocean via the so-called Agulhas Leakage (AL), thereby influencing the variability of the Atlantic meridional overturning circulation (hereafter AMOC; Peeters et al., 2004; Beal et al., 2011; Biastoch et al., 2009, 2015). Paleoclimate studies have pointed to the importance of the AL on glacial-interglacial time scales and suggested a vital role of the AL in steering the AMOC variability on millennial time scales (Peeters et al., 2004; Simon et al., 2013). The interocean exchange of heat and salt via the AL is dynamically excited through the mean flow and vigorous mesoscale ocean eddies and filaments, which are shed into the southern Atlantic at the Agulhas Retroflection (AR; **Fig. 1**; Biastoch et al., 2015). The sea surface temperature (SST) and salinity (SSS) variability in the AC, which feeds the AL, are suggested to be related to upstream wind and current variability in the southern Indian Ocean (Backeberg et al., 2010; Biastoch et al., 2008, 2009, 2015; Rouault et al., 2009). There is evidence from satellite altimetry observations that mesoscale variability upstream of the AC has strengthened since 1993, resulting in accelerated eddy propagation into the AC and AR regions (Backeberg, 2012). This is related to enhanced ocean current transport in response to an increase in wind stress curl in the southern Indian Ocean trade winds (Backeberg et al., 2012). AC and southern Mozambique Channel historical SST have increased since the early 1980s, even though sensible and latent heat flux from the ocean to the atmosphere also increased and consequently should have cooled the surface ocean (McClanahan et al., 2008; Rouault et al., 2009). High-resolution regional ocean modelling, paired with observational estimates, suggest that increased oceanic heat advection by large-scale currents in the southern Indian Ocean, has contributed to the increase in SST of the AC, thereby largely offsetting the turbulent heat flux (Rouault et al., 2009). Recently, it has been suggested that current rapid Indian Ocean warming can play a role in sustaining the AMOC through remote impacts on the Walker Circulation, thereby affecting surface salinity in the tropical Atlantic (Hu & Fedorov, 2019). Indian Ocean warming and strengthening of the AMOC are shown to also operate in models driven by future greenhouse gas ($CO_2$) emission increases (Hu & Fedorov, 2019). However, large uncertainties remain concerning the role of large-scale atmospheric circulation (wind and sea level pressure) on SST and SSS over longer time scales and their influence on AC SST and SSS variability due to short observational datasets and the highly dynamic nature of the region (Rouault et al., 2009; Backeberg et al., 2010).

On interannual timescales, the AC and AL region is apparently sensitive to the El Nino-Southern Oscillation (ENSO) with a 24-month lag (Putrasahan et al., 2016; Elipot & Beal, 2018; Paris et al., 2018; Trott et al., 2021). Anomalous wind stress curl in the southern Indian Ocean in response to ENSO excites the westward propagating oceanic Rossby waves (Feng & Myers, 2003; Palastanga et al., 2006; Grunsreich et al., 2011). These Rossby waves, in turn, modulate the transport of heat and

salinity anomalies across a tropical (12°S) and subtropical (25°S) pathway (Putrasahan et al., 2016). Concomitant El Niño and positive Indian Ocean dipole (IOD) events lead to freshening in the equatorial Indian Ocean and saltier anomalies off Sumatra and vice versa for La Niña and negative IOD events (Grunsreich et al., 2011). Although Grunsreich et al. (2011) argue that freshening during El Niño and positive IOD may reach the far southwestern Indian Ocean, their study used salinity from SODA reanalysis since 1870, which is currently not validated against other salinity products or palaeoclimate reconstructions (Giese & Ray, 2011). Rainfall variability over southwestern Madagascar, including Toliara (near the coral reef location), showed strong negative correlations with ENSO (r=-0.61, 99% significant; Randriamahefasoa & Reason, 2017). Between 1970 and 2000 rainfall was below average in five out of seven El Niño events for Toliara while four out of six La Niña events received average or above average rainfall. Correlations of Toliara rainfall with global SST revealed an ENSO-like spatial pattern across the Indo-Pacific and a subtropical Indian Ocean Dipole SST pattern east of Madagascar (warm SST south-southeast of Madagascar, cold northeast of Madagascar and central Indian Ocean SST; Randriamahefasoa & Reason, 2017). El Niño events lead to thermal stress in the southern Mozambique Channel with coral bleaching in coral reefs of Ifaty and Toliara inferred for 1998 (not reported) and observed in 2016 (Zinke et al., 2004; McClanahan et al., 2009; Harris et al., 2010; Obura et al., 2017). Thus, ENSO appears to have a significant influence on temperature and rainfall variability in the southern Mozambique Channel.

Paleoclimate reconstructions based on a 334 year, annually resolved coral proxy record from Ifaty reef off southwestern Madagascar in the southern Mozambique Channel (MC) also point to significant interdecadal SST variations in the greater AC core region, with tight teleconnections to the mid-latitude South Atlantic and South Indian Ocean SST (Zinke et al., 2004, 2014; Bruggemann et al., 2012). Numerical modelling exhibited that this coral-derived SST record was representative of the AC's wider core region since 1958 on interannual to decadal time scales (Zinke et al., 2014). This coral SST record revealed a strong relationship between local southern MC SST and ENSO (when ENSO variability was strong) and suggests a link with the Pacific Decadal Oscillation (Zinke et al., 2004; Crueger et al., 2009). However, the coral-based reconstruction revealed solely the SST variability of the AC core region, while long-term salinity reconstructions at high temporal resolution are lacking. SSS has emerged as an excellent indicator for the global hydrological cycle and its distribution across the oceans by currents (Hegerl et al., 2015; Skliris et al., 2014). This is because salinity provides an integrative measure of ocean advection, precipitation-evaporation (P-E), and ocean density changes (Skliris et al., 2014). Measurements of the $\delta^{18}O$ in seawater (hereafter $\delta^{18}O_{seawater}$), which is strongly related to P-E as well as horizontal and vertical ocean transport processes affecting salinity, allow for assessing large-scale and regional hydrological changes driven by climate variability (Le Grande & Schmidt, 2006). However, salinity and $\delta^{18}O_{seawater}$ data are still scarce in many parts of our oceans, including the AC and AR region, and salinity measurements with global coverage only recently began in 2010 using satellites (Boutin et al., 2021)). Salinity variability can be estimated from reanalysis products such as SODA (Giese & Ray, 2011) or observation-based products such as the EN4 (Good et al., 2013) back to 1958, yet their suitability has not been tested over long time periods. This hampers a full understanding of thermohaline circulation variability transmitted via the AC system from the preindustrial into the current warm period.

Here, we present an annually resolved coral skeletal Sr/Ca composite record paired with an established composite oxygen isotope record from Ifaty and Tulear reefs in the southern MC southwest of Madagascar, to obtain a 334 year-long reconstruction of $\delta^{18}O_{seawater}$ changes related to surface salinity variability in the wider southern MC and the AC core region.

We compare the coral Sr/Ca and $\delta^{18}O_{seawater}$ records with reanalysis and observation-based products of SST (ERSST5 and HadISST1; Huang et al., 2017; Rayner et al., 2003) and SSS (SODA 2.1.6; Giese & Rayner, 2011) to show that the coral records trace the local interannual to decadal SST and SSS variability. Furthermore, we use these reanalysis and observation-based products as well as a hindcast (1958-2018) simulation with the ocean/sea-ice model configuration INALT20 at mesoscale eddy rich resolution (Schwarzkopf et al., 2019) to assess the relationship of temperature and salinity variability at

the coral site with that of the greater AC region. From this, in combination with historical wind stress observations from ICOADS (Woodruff et al., 2011) and coral records from other sites in the Indian Ocean, we show that it is possible to infer relationships of the regional SST and SSS variability with large-scale Indian Ocean variability, which ultimately may influence the surface thermohaline circulation (Rahmstorf et al., 2015).

**2 Methods and Materials**

**2.1 Coral core collection and sampling**

Coral cores from massive *Porites* sp. at Ifaty and Tulear reefs were collected in October 1995 during the European Union TESTREEF program from the Ifaty-Ranobé lagoon and the Great Barrier of Tulear (southwest Madagascar; Zinke et al., 2004). The Ifaty and Tulear coral reef sites are described in detail in Zinke et al. (2004). Core Ifaty-4 (4.06m length), core Ifaty-1

(1.93m length), and Tulear-3 (1.80m length) were obtained from a depth of 1.1m, 1.8m, and 0.6m below mean tide level. The average growth rate of core Ifaty-4 was 0.99±0.15cm per year, whereas Ifaty-1 averaged 1.28±0.24cm and Tulear-3 averaged 1.54±0.25cm per year.

All cores were sectioned to a thickness of 7 mm, and slabs were cleaned in 10% hydrogen peroxide for 48 h to remove organic matter at GEOMAR Kiel. Slabs were subsequently rinsed several times with demineralized water and dried with compressed

air. For complete removal of any moisture within the coral skeleton, the sample was placed in an oven for 24 h at 40 °C. Finally, the slabs were X-rayed to reveal annual density banding (Zinke et al., 2004).

A high-resolution profile for stable isotope analysis on core Ifaty-4 was drilled using a computer-controlled drilling device along the growth axis as observed in X-ray-radiograph-positive prints (Zinke et al., 2004). Subsamples were drilled at a distance of 1 mm for the years 1995–1920 and 2 mm for the older part of the core; the drilling depth was 3 mm using a 0.5

mm dental drill at 1000 rpm. The 1 to 2 mm sample spacing provides approximately monthly or bi-monthly resolution for $\delta^{18}O$ (Sr/Ca for several multidecadal periods; see Zinke et al., 2004), respectively. We resampled the Ifaty-4 core at annual resolution for Sr/Ca, except for multidecadal periods subsampled previously at bimonthly resolution (Zinke et al., 2004) following the established and precise age model of the high-resolution $\delta^{18}O$ sampling from austral summer to summer in any given annual

cycle (Tab. S1). Cores Ifaty-1 and Tular-3 were sampled at annual resolution along the major growth axis following the density

pattern from summer to summer in any given annual cycle, established from X-ray-radiograph-positive prints.

## 2.2. Analytical procedures of $\delta^{18}O$ and Sr/Ca

The high-resolution samples of core Ifaty-4 were reacted with 100% $H_3PO_4$ at 75 °C in an automated carbonate reaction device (Kiel Device) connected to a Finnigan MAT 252 mass spectrometer (University Erlangen). Average precision based on

duplicate sample analysis and on multiple analysis of NBS 19 is ±0.07‰ for $\delta^{18}O$ (1σ). The annual samples for cores Ifaty-1 and Tular-3 were reacted with 100% $H_3PO_4$ at 75 °C in an automated carbonate reaction device (Kiel Device) connected to a Finnigan MAT 252 mass spectrometer at the VU University of Amsterdam. Average precision based on duplicate sample analysis and on multiple analysis of NBS 19 is ±0.08‰ for $\delta^{18}O$ (1σ).

Sr/Ca ratios were measured at the University of Kiel with a simultaneous inductively coupled plasma optical emission

spectrometer (ICP-OES, Spectro Ciros CCD SOP), following a combination of the techniques described by Schrag[77] and de Velliers[78]. Sr and Ca were measured at their 421 and 317 nm emission lines, respectively. 175±25 μg of coral powder was dissolved in 1 ml nitric acid ($HNO_3$ 2%). Prior to analysis, this solution was further diluted with 4 ml $HNO_3$ 2% to a final concentration of approximately 8 ppm. An analogously prepared in-house standard (Mayotte coral) was measured after each sample batch of 6 samples to correct for drift effects. The international reference material JCp-1 (coral powder) was analysed

at the beginning and end of every measurement run. Internal analytical precision based on replicate Sr/Ca measurements was 0.008 mmol/mol (1σ) or 0.08 %. The average Sr/Ca value of the JCp-1 standard from multiple measurements on the same day and on consecutive days was 8.831 mmol/mol with 0.085 % relative standard deviation (RSD). Comparison to the certified Sr/Ca value of 8.838 mmol/mol[79] with an expanded uncertainty of 0.089 mmol/mol indicates a high external precision of <0.08 %.


## 2.3 Age model uncertainty

We used the already published (bi)monthly resolved Ifaty-4 coral $\delta^{18}O$ time series from 1660 to 1994 (Zinke et al., 2004). The high-resolution Ifaty-4 coral $\delta^{18}O$ record enabled us to compute a precise coral $\delta^{18}O$ annual chronology, averaged between March to February. We used the Ifaty-4 core as our best-dated reference time series to ensure that the yearly sampled

chronologies of Ifaty-1 and Tular-3 aligned well.

We have estimated the uncertainty in annual mean Sr/Ca, $\delta^{18}O$ and $\delta^{18}O_{sw}$ due to potential sampling errors with a pseudoproxy approach (**Figs. S1-S2**). The pseudo annual means were calculated from "year's start" to "year's end". The year's start and end were randomly selected from December to May (March ± 2~3 months) and from December to May (February ± 2~3 months). We looped this estimate in 20,000 times and took the median of the looped results as the pseudo annual mean. The pseudo

annual mean was compared with the true mean value (mean from March to February). The difference of Sr/Ca between the

true and pseudo values is 0.003 ± 0.007 mmol/mol (1σ) (i.e., about ± 0.1 ºC) while the difference in $\delta^{18}O$ is 0.02 ± 0.014‰. Because of SST-related seasonality, Sr/Ca and $\delta^{18}O$ may have a bias towards positive values (lower SST), however this bias is low. $\delta^{18}O_{sw}$ estimated from paired coral $\delta^{18}O$ and Sr/Ca measurements (see section 2.4 for methodology) is not significantly affected by the age model error (0.00±0.03‰ between true and pseudo values).


## 2.4. SST and $\delta^{18}O_{seawater}$ reconstruction

The composite SST and $\delta^{18}O_{seawater}$ (hereafter $\delta^{18}O_{sw}$) records and their uncertainty envelopes were estimated from medians and percentiles of their simulated distributions using a Monte Carlo approach developed by Watanabe & Pfeiffer (2022). The method was expanded to include the intercolonial differences of coral $\delta^{18}O$ and Sr/Ca in the uncertainty estimates. In a first
step, SST records were estimated from Sr/Ca and coral $\delta^{18}O$ using the centering method of Cahyarini et al. (2008). Variations in SST and $\delta^{18}O_{sw}$ (hereafter $SST_{center}i$; $\delta^{18}O_{sw-center}i$) are estimated from coral Sr/Ca and $\delta^{18}O$ (hereafter $\delta^{18}O_c$) centered to their 1961-1990 mean, based on the assumption that coral Sr/Ca is solely a function of SST and that coral $\delta^{18}O$ is a function of both SST and oxygen isotopic composition of the seawater ($\delta^{18}O_{sw}$). Effects of $\delta^{18}O_{sw}$ on coral $\delta^{18}O$ are separated from thermal effects by subtracting the temperature component derived from Sr/Ca from the $\delta^{18}O$ in the coral skeleton.

$$SST_{center} = (Sr/Ca - \overline{Sr/Ca})/\beta$$
$$\delta^{18}O_{sw-center} = (\delta^{18}O_c - \overline{\delta^{18}O_c}) - (b/\beta) \times (Sr/Ca - \overline{Sr/Ca})$$
, where $Sr/Ca_{center}$ and $\delta^{18}O_{c-center}$ are centered records of Sr/Ca and $\delta^{18}O_c$ by removing the 1961-1990 mean, and $\beta$ and b are the calibration slopes of Sr/Ca-SST and $\delta^{18}O_{SST}$-SST, respectively.

These equations were rewritten for centered records ($Sr/Ca_{center}$ and $\delta^{18}O_{c-center}$, removing the 1961-1990 mean) and their
uncertainties. The latter include intercolonial differences of Sr/Ca and $\delta^{18}O_c$ ($D_{Sr/Ca}$ and $D_{\delta 18Oc}$), the analytical uncertainties of the proxy analyses ($E_{Sr/Ca}i$ and $E_{\delta 18Oc}i$) and the slope uncertainties of the proxy-SST relationships ($E_\beta i$ and $E_b i$). The equations are as follows:

$$SST_{center}i = (Sr/Ca_{center} \pm D_{Sr/Ca}i \pm E_{Sr/Ca}i)/(\beta \pm E_\beta i)$$
$$\delta^{18}O_{sw-center}i = (\delta^{18}O_{c-center} + D_{\delta 18Oc}i \pm E_{\delta 18Oc}i) - (b \pm E_b i)/(\beta \pm E_\beta i) \times (Sr/Ca_{center} \pm D_{Sr/Ca}i \pm E_{Sr/Ca}i)$$


We calculated $SST_{center}i$ and $\delta^{18}O_{sw-center}i$ in a loop 20,000 times by adding random values following the normal distributions derived from 1σ of slope errors and analytical errors. Uncertainties deriving from intercolonial differences ($D_{Sr/Ca}$ and $D_{\delta 18Oc}$) are estimated from the standard deviation observed between the three cores covering 1994 to 1905 (±0.04 mmol/mol, 1σ; ±0.10‰$_{VPDB}$). For the time period covered by one single core only (1881~1661 CE), $D_{Sr/Ca}$ and $D_{\delta 18Oc}$ were
added as random values (from a normal distribution with a mean of 0) on $Sr/Ca_{center}$ and $\delta^{18}O_{c-center}$ to indicate potential uncertainties arising from intercolonial differences. The analytical uncertainty of $\delta^{18}O_{sw}$ ($\sigma_{\delta 18Osw}$) in this study is 0.103‰ for bimonthly values, which reduces to 0.058‰ for annual means computed from bimonthly data according to the formula $\sigma_{total} = (2/N)^{1/2}$ (Bevington, 1969). The Sr/Ca-SST slope and its uncertainty ($\beta \pm E_\beta i$) is -0.06 ± 0.01 mmol/mol/ºC (1σ, Corrège, 2006) and the $\delta^{18}O$-SST slope and its uncertainty (b $\pm E_b i$) is -0.22 ± 0.02 ‰/ºC (1σ, Lough, 2004; Thompson et al., 2011).
The calculation of composite SST and $\delta^{18}O_{sw}$ uncertainty (beyond analytical uncertainty) consists of the following steps: 1) all proxy records are centered by removing the 1961-1990 mean, 2) a pair of Sr/Ca and $\delta^{18}O$ is randomly selected in each year from the three coral core datasets, 3) Monte Carlo parameters are calculated by adding random values on the proxy-

SST slopes, Sr/Ca, and $\delta^{18}O$ (random values are normally distributed numbers in the 1 σ range of slope errors and analytical errors, respectively), 4) $\delta^{18}O_{sw}$ is calculated using the Monte Carlo parameters, 5) step 2. to 4. is repeated in a loop 20,000 times, and 6) the median and percentiles are estimated from the resulting distributions. Uncertainty envelopes for the single record (1881~1661 CE) were calculated using the same procedures used for the composite record, omitting the second step. Anomalies of SST and $\delta^{18}O_{sw}$ from all annual coral records (March to February) were reported as centered records relative to the 1961-1990 mean.

## 2.3. Observational data and simulation with the ocean/sea-ice model configuration INALT20

To show that the SSS and SST variability at Ifaty are representative for the wider AC region, we compared different reanalysis and observation-based products, as well as our coral data, and analysed the interannual to decadal temperature and salinity variability in a mesoscale eddy-rich ocean model simulation. For SST, we selected ERSSTv5 (1854 to 1995; Huang et al., 2017) and HadISST1 (1870 to 1995; Rayner et al., 2008), both grounded on the ICOADS dataset (Woodruff et al., 2011). Gridded SST products may suffer from high uncertainties pre-1970, when observational coverage decreases and uncertainties become large (**Fig. S3**). Nevertheless, the ERSSTv5 and HadISST1 data provide an estimate for likely SST change since the 19[th] century. For salinity, we utilized Simple Ocean Data Assimilation (SODA 2.1.6, 1958 to 2008; Giese & Rayner, 2011). Salinity from the EN4 dataset (Version 4.2; Good et al., 2013) was not used because subtropical Indian Ocean locations showed limited data coverage and no significant correlation with SODA salinity (Tab. S2). The employed model simulation is a hindcast (1958-2018, experiment identifier INALT20.L46-KFS10X, here referred to as INALT20-JRA) with the global ocean/sea-ice model configuration INALT20, which is part of the INALT family introduced in Schwarzkopf et al. (2019). INALT20 has a global resolution of ¼° that is regionally (63°S–10°N and 70°W–70°E) refined to 1/20°, to resolve the intricate mesoscale circulation features in the extended AC region and their impact on the South Atlantic. Note though that the employed hindcast simulation differs from the one described in Schwarzkopf et al. (2019) as detailed in Schmidt et al. (2021) and Biastoch et al. (2021). In particular, it was run under the novel JRA55-do atmospheric forcing (Tsujino et al. 2018), which is available at higher resolution and for a longer time period than the previously employed COREv2 data set (Large and Yeager, 2009). From the model simulation, instead of SST and SSS, we used near-surface salinity (NSS) and near-surface temperature (NST) taken at the vertical model level 3 (16,36 m) to avoid a direct restriction to the surface forcing (see Biastoch et al., 2015 for the rationale). Salinity and temperature timeseries were extracted from observation-based products and simulations for Ifaty (43°E and 23°S) and a representative location within the AC (30°E and 32°S). Correlations were calculated using annual means (Jan-Dec for comparison with the model, Mar-Feb for comparisons with the coral). All Figures show annual anomalies relative to the 1961 to 1990 mean.

We utilize historical observations from ICOADS wind stress (1850 to 2010; Freeman et al., 2017) to infer large-scale variability in the atmospheric circulation over the Indian Ocean (10-40°S, 50-100°E) and its potential relation to ocean heat (SST) and salt ($\delta^{18}O_{seawater}$) zonal advective transport and their relationship to large-scale (10-40°S, 50-100°E) Indian Ocean

variability. 20[th]-century reanalysis data for the precipitation-evaporation (P-E) balance were employed to assess long-term relationships with freshwater flux over oceanic areas (Giese et al., 2016). The Multivariate El Niño-Southern Oscillation (ENSO) index (MEI) was used to assess interannual variability (Wolter & Timlin, 1998, 2011). All gridded datasets for the study area were extracted as annual anomalies relative to the 1961 to 1990 mean using the KNMI Climate Explorer (Trouet & Oldenborgh, 2013).

## 3 Results

### 3.1 Validation of reconstructed SST and $\delta^{18}O_{seawater}$ at Ifaty

The new $\delta^{18}O_{seawater}$ reconstruction is based on three *Porites* paired Sr/Ca, and coral oxygen isotope ($\delta^{18}O$) records at annual resolution from Ifaty and Tulear coral reefs off southwestern Madagascar (43°E, 23°S), covering the past 334 years (**Fig. 2**; Zinke et al., 2014). The composite annual chronology extends from 1661 to 1994, with cores Ifaty-4, Ifaty-1, and Tulear-3 covering the years 1661-1994, 1890-1994, and 1905-1994, respectively. The uncertainty estimates in $\delta^{18}O$ and Sr/Ca were derived from the replicated time period (**Fig. 2d**, see Methods section).

Our new mean annual Sr/Ca-SST record largely covaries with the established $\delta^{18}O$-SST, yet shows a higher amplitude variability (**Figs. 2a, b and Fig. S1**). However, especially in the 18[th] and 19[th] centuries, lower mean SST in Sr/Ca-SST than $\delta^{18}O$-SST results in lower $\delta^{18}O_{seawater}$ anomalies and vice versa for cooler mean SST in Sr/Ca-SST. Between 1661 and 1995, Sr/Ca-SST records a linear warming trend of 0.94±0.26 ºC, while $\delta^{18}O$-SST indicates an increase of 0.83±0.21 ºC, both statistically significant at p<0.001 (Zinke et al., 2004). Detrended annual coral $\delta^{18}O$-SST and Sr/Ca-SST are significantly correlated with detrended annual mean (March to February) SST of instrumental records at Ifaty (**Tab. S2**). The new $\delta^{18}O_{seawater}$ reconstruction displays no linear trend and is dominated by multidecadal to centennial variability throughout the 334-year record punctuated by strong interdecadal and interannual variability (**Fig. 2c**). Large amplitude interannual and decadal variability in $\delta^{18}O_{seawater}$ is observed during the Late Maunder Minimum (hereafter LMM; 1670-1710). The late 18[th], most of the 19[th] century, and individual years during the LMM show the lowest $\delta^{18}O_{seawater}$ anomalies, thus low saline surface water conditions. The most saline conditions are observed for several years during the LMM and the late 19[th] century. The early to mid 20[th] century (1910-1940) is characterized by high saline surface waters  and a freshening is recorded over the period 1958 to 1995.

We validated the mean annual Ifaty-Tulear coral $\delta^{18}O_{seawater}$ reconstruction with surface salinity data from SODA 2.1.6, available since 1958 (Giese & Ray, 2011, Tab. 1; **Fig. 2c**), and by using 1° gridded HadISST (Rayner et al., 2003) and 2° gridded ERSST5 instead of Sr/Ca to reconstruct $\delta^{18}O_{seawater}$ since 1854 (Huang et al., 2017; **Figs. 3 and S5**). Salinity from SODA reanalysis from a grid near Ifaty reef (43°E, 23°S) were used. Our detrended $\delta^{18}O_{seawater}$ record based on Sr/Ca shows statistically significant correlations with Ifaty SODA salinity (r= 0.50, p<0.05, N= 36; **Tab. 1; Fig. 3a-d**), assuming 18 degrees of freedom (taking into account autocorrelation in SSS data). The best line of fit between annual mean $\delta^{18}O_{seawater}$ and SODA

salinity has a slope of 0.58±0.06 ‰/psu, close the observed mean slopes of subtropical waters between 0.4 and 0.6 ‰/psu (LeGrande and Schmidt, 2006, 2011). A freshening trend is observed in both SODA salinity (-0.06±0.01psu per decade; p=0.001) and $\delta^{18}O_{seawater}$ (-0.06±0.02‰ per decade; p<0.001) in the overlapping period of both records between 1958 and 1995.

SODA salinity suggests a switch to more saline conditions after 2000 (**Figs. 2c and 3a**). $\delta^{18}O_{seawater}$ reconstructed from HadISST and ERSST5 co-vary with Sr/Ca-based $\delta^{18}O_{seawater}$ between 1854 and 1995, both showing lowest saline conditions on record in the late 19$^{th}$ century and between 1958 and 1995 (**Fig. S5**). The salinity from SODA over the wider Indian Ocean along the path of the South Equatorial Current and through the Mozambique Channel largely covaries with salinity at Ifaty since 1958 (**Fig. S6**). Ifaty-Tulear $\delta^{18}O_{seawater}$ anomalies and SODA salinity together with Sr/Ca-SST anomalies and observed

ERSST5 at Ifaty indicate a warming and freshening tendency between 1970 and 2000 (**Fig. S7**). For the record between 1854 and 1995, it appears as if decreasing (increasing) Ifaty-Tulear $\delta^{18}O_{seawater}$, i.e., freshening (salinification), coincides with decreasing (increasing) Sr/Ca-SST and ERSST5, i.e., cooling (warming). Yet, the relationship is weak and interannual to decadal variability is not statistically significantly correlated. Hence, no robust correlation or causality could be established between the temporal evolution of regional temperature and salinity.

The (detrended) year-to-year NST/SST variability at Ifaty in the hindcast simulation with the mesoscale eddy-rich ocean/sea-ice model configuration INALT20 (Schwarzkopf et al., 2019) is significantly correlated (95% confidence level, t-test with effective degrees of freedom determined via e-folding scale of autocorrelation function) with ERSST5 (r= 0.62, p<0.001; **Fig. 4a-c**) and HadISST (1958-2018, r=0.37, p<0.001). Detrended NST/SST is not significantly correlated between the simulation and the coral Sr/Ca-SST record (r=0.24, p=0.16; **Fig. 4a**). The (detrended) year-to-year NSS/SSS variability at

Ifaty in the hindcast simulation is not significantly correlated with SODA (r=0.04, p=0.45, **Fig. 4d-f**) yet with EN4 (r=0.45, p=0.045, not shown) at Ifaty. Detrended NSS/SSS at Ifaty in the hindcast simulation is also not significantly correlated with the reconstructed coral $\delta^{18}O_{seawater}$ record (**Fig. 4d-f**). The model simulation further reveals no clear, direct relationship between NST/NSS and net surface heat/freshwater fluxes at Ifaty (**Fig. 5**)

**3.2. Representativeness of SST and SSS variability at Ifaty for variability in the wider Agulhas Current region**

To further validate our hypothesis that the Sr/Ca and $\delta^{18}O_{seawater}$ records from the Ifaty-Tulear reef complex are representative for temperature and salinity in the wider AC region, we analysed the relationship between the temporal evolution of annual mean salinity and temperature at Ifaty (centered at 43°E, 23°S) and within the AC core region (centered at 30°E, 32°S) in observations (ERSST5 and HadISST; Rayner et al., 2003; Huang et al., 2017), a hindcast simulation with the

mesoscale eddy-rich ocean/sea-ice model configuration INALT20 (Schwarzkopf et al., 2019), as well as in SODA reanalysis-based products (Giese & Rayner, 2011).

Detrended Ifaty ERSST5 (and HadISST1) are significantly correlated with AC ERSST5 (1870-1995; r=0.74, p<0.001). Detrended annual coral $\delta^{18}O$-SST and Sr/Ca-SST are significantly correlated with detrended annual mean (March to February) SST of instrumental records within the AC core region (**Tab. S2**). Our detrended $\delta^{18}O_{seawater}$ record based on

Sr/Ca shows statistically significant correlations with AC core region salinity (r= 0.57, p<0.001, N= 36; **Tab. 1**), assuming 18 degrees of freedom (taking into account autocorrelation in SSS data). Spatial correlations between Ifaty-Tulear detrended $\delta^{18}O_{seawater}$ and SODA SSS reveals a significant relationship across the greater Agulhas region (**Fig. 3b**). For the hindcast simulation as well as for all data products individually, the interannual to decadal variability of NST/SST and NSS/SSS at Ifaty indeed seem representative for NST/SST and NSS/SSS at the chosen location in the AC (**Fig. 4**), as well as for that of

the wider AC region (not shown), even though the exact year-to-year SSS/NSS variability between the individual records differs (**Fig. 4**). The (detrended) year-to-year NST/SST variability between Ifaty and the AC are significantly correlated (95% confidence level, t-test with effective degrees of freedom determined via e-folding scale of autocorrelation function) in the simulation (1958-2018, r=0.60, p <0.001, **Fig. 4b**), as well as with HadISST (1958-2018, r=0.72, p<0.001, **Fig. 4c**) and SODA (1958-2008, r=0.66, p=0.005, not shown).

Detrended Ifaty and AC SODA SSS are significantly correlated (1958-1995; r=0.82, p=0.003) However, the (detrended) year-to-year NSS/SSS variability between Ifaty and the AC (1958-present) are significantly correlated in the simulation (r=0.75, p<0.001, **Fig. 4e**), yet not significant with the coral $\delta^{18}O_{seawater}$ record. The latter may be a result of the shorter timeseries (the here considered overlapping time period of the simulation and data products with the coral record is 1958-1994). Even though the relationship between AC and Ifaty salinity variability is robust in the simulation, there is no

agreement on the actual temporal evolution of salinity since 1958 (**Fig. 4d**). There is no considerable correlation of (detrended) year-to-year NSS/SSS variability between the simulation and data products, among the data products, and between the simulation or data products and the coral $\delta^{18}O_{seawater}$ record (**Fig. 4d**). The only exception is SODA salinity, which shows a temporal evolution that agrees well with that of the coral $\delta^{18}O_{seawater}$ record.

The model simulation further reveals no clear, direct relationship between NST/NSS and net surface heat/freshwater

fluxes at IFA and AC regions (**Fig. 5**), and hence supports the idea that NST/NSS variability at IFA and in the wider Agulhas region is dominated by oceanic processes such as advective heat and freshwater/salt transports.

### 3.3. Large-scale drivers of reconstructed $\delta^{18}O_{seawater}$

The question arises as to what is driving long-term changes and interannual to decadal variability in oceanic heat and

freshwater/salt transport in the greater AC core region. First, we assessed if regional rainfall or freshwater discharge from land is driving salinity anomalies at Ifaty and Tulear reefs. We utilized rainfall station data from a weather station at Tulear in southwestern Madagascar dating back to 1951, but with several gaps in the 1990s (**Fig. S8**). Rainfall and salinity or $\delta^{18}O_{seawater}$ should be negatively correlated when rainfall or freshwater runoff influences the signal. Tulear rainfall reveals a positive correlation with both Ifaty and AC salinity data and reconstructed $\delta^{18}O_{seawater}$ between 1951 and 1994. Thus, rainfall over

adjacent land along the latitudinal band of the 23°S in Madagascar is weakly positively correlated with salinity and $\delta^{18}O_{seawater}$. This implies that regional rainfall on land is not a driver of ocean salinity.

We utilized historical observations from ICOADS (Freeman et al., 2017) wind stress to infer changes in the large-

scale atmospheric circulation over the Indian Ocean and their potential impact on oceanic heat and salt transport. Negative anomalies in zonal wind stress translate into a strengthening of easterly anomalies across the southern Indian Ocean along the South Equatorial Current (SEC) which would invigorate zonal ocean transport of SST and salinity anomalies. ICOADS zonal wind stress averaged over the southern Indian Ocean (10-40°S, 50-100°E) east of Madagascar is available since 1855, with significant data gaps between 1880 and the 1940s. It indicates a trend to more negative anomalies since 1947, where data are complete (**Fig. 6**). Historical wind data from ICOADS (Freeman et al., 2017) indicate that this trend might have been part of a long-term decrease since the mid-19[th] century. ERA 20[th] century reanalysis data also indicate a trend to more negative zonal wind stress anomalies between 1900 and the present (not shown). ICOADS zonal wind stress averaged over the southern Indian Ocean (10-40°S, 50-100°E) shows a positive correlation (r= 0.73, p<0.001, N= 81) with zonal wind stress south of Madagascar in the greater AC region (20-40°S, 30-45°E), which also holds after detrending (r= 0.60, p<0.001). Our reconstructed annual $\delta^{18}O_{seawater}$ record indicates a positive correlation (r= 0.55, p<0.001) with southern Indian Ocean (10-40°S, 50-100°E) ICOADS zonal wind stress, which also holds after detrending (r= 0.41, p=0.001; **Fig. 6**). We low-pass filtered (5-year LOESS filter) the ICOADS wind stress data and our $\delta^{18}O_{seawater}$ reconstruction to assess potential relationships also on sub- decadal time scales. Our low-pass filtered reconstructed $\delta^{18}O_{seawater}$ record indicates a positive correlation (r= 0.67, p= 0.0063) with the southern Indian Ocean (10-40°S, 50-100°E) ICOADS zonal wind stress, pointing to easterly wind anomalies driving ocean advection of the salinity signal across southern Indian Ocean.

Furthermore, spatial correlations with 20[th]-century reanalysis data (Giese et al., 2016) for the P-E balance were chosen to assess long-term relationships with freshwater flux over oceanic areas (**Fig. 7**). Our results reveal that Ifaty salinity and reconstructed $\delta^{18}O_{seawater}$ is related to P-E over the southern Indian Ocean, stretching meridionally from the northwest to southeast (east of Madagascar) and the northern Mozambique Channel (MC; **Figs. 7a, b**). Lagged correlations indicate that the P-E anomalies are propagated via Rossby waves into the MC and the AC region within 2 years after the initial P-E anomaly east of Madagascar (**Fig. 7c**). Since variability in the propagation of salinity anomalies is known to be driven by ENSO (Putrasahan et al., 2016), we assessed lagged correlations between the MEI index with the AC region 20[th]-century reanalysis P-E record, SODA salinity, and Ifaty reconstructed $\delta^{18}O_{seawater}$ between 1950 and 2016 and 1950 and 1994, respectively (**Figs. 7c-d; Fig. S9**). Lagged correlations indicate a freshening of surface waters 12 to 18 months after El Niño and salinification after La Niña (**Figs. 7c-f; Fig. S9**). Results were similar using the observational and palaeo-Niño3.4 indices instead of the MEI index between 1880 and 1994 and 1750 to 1994 (**Figs. 7g-h; Fig. S10**). The southern MC P-E balance, on the contrary, is negatively correlated at six to twelve-month lag with the MEI, implying lower precipitation over the ocean following El Niño (**Fig. 7c**). Thus, lower (higher) salinity in the southern MC following El Niño (La Niña) implies a larger role of ocean Rossby wave propagation over the regional P-E balance in interannual salinity variability.

We applied spectral analysis to test for the presence of interannual frequency bands and the 2-year timescale associated with propagation of Rossby waves identified by our lagged correlation analysis. Spectral analysis with the Multitaper method revealed that interannual variability in reconstructed SST and $\delta^{18}O_{seawater}$ is dominated by frequencies

ranging from two to four years (**Fig. 8**). Similar frequencies were also found in observed ERSST5 and SODA SSS data for Ifaty and the AC region (**Fig. S11**). Such timescales of variability are typical for ENSO interannual frequency bands. Both the two and four-year frequencies were of the highest magnitude between 1661 and 1900 and generally lower in the 20th century. The mid- and late 20th century indicate higher interannual variability than early 20th century (**Fig. 8**). We also assessed the

variability in reconstructed SST and $\delta^{18}O_{seawater}$ through moving 30-year standard deviations, stepped by five years, following Abram et al. (2020). This analysis also reveals highest variability in reconstructed SST and $\delta^{18}O_{seawater}$ between 1661 and 1900, and diminished variability thereafter (**Fig. 9**). Blackman-Tuckey spectral coherence analysis suggests that Ifaty-Tulear Sr/Ca-SST and $\delta^{18}O_{seawater}$ is coherent with palaeo-ENSO reconstructions of Emile-Geay et al., (2013; EG13) and Steiger et al., (2018; PHYDA) at frequencies between 2 and 4 years, as well as decadal bands ranging between 8-16 years and >30 years (**Fig. S12**).

Cross-wavelet coherence analysis (Torrence and Compo, 1998) revealed strongest coherence in the 2-4 years frequency band with EG13 between 1650 and 1850, around 1900 and from the 1940's onwards **Fig. 9b**). The PHYDA ENSO reconstruction indicates coherence in the 2 to 4 years frequency band wth Ifaty-Tulear $\delta^{18}O_{seawater}$ similar to EG13 between 1850 and 1995, yet weaker in the 18th century. (**Fig.9c**). The 8-16 years frequency band shows high coherence between Ifaty-Tulear $\delta^{18}O_{seawater}$ and all ENSO records in the 20th century, between 1780 and 1870 and 1650 to 1710 (**Figs. 9b-d**). Wavelet coherence between

Ifaty-Tulear Sr/Ca-SST and ENSO revealed largely similar coherent periods as $\delta^{18}O_{seawater}$ (**Fig. S13**). We band-pass filtered the $\delta^{18}O_{seawater}$ reconstruction and palaeo-ENSO reconstructions for interannual to multi-decadal periodicities and computed running correlations to further assess long-term relationships (**Fig. 10; Fig. S14**; Emile-Geay et al., 2013; Steiger et al., 2018). The palaeo-ENSO reconstructions do not agree with each other for large parts of the record since 1661 in various frequency bands (**Fig. 10**). The best agreement is found for the period where both ENSO reconstructions were calibrated with instrumental

data (1870-1995) for the 3.3 to 4 years frequency band (**Fig. 10**). Consequently, our band pass filtered $\delta^{18}O_{seawater}$ record shows various levels of agreement and disagreement (or phase lags) with individual ENSO reconstructions. However, these results support the results from cross-wavelet coherence analysis indicating varying phase lags between ENSO records and Ifaty-Tulear $\delta^{18}O_{seawater}$ on decadal to interdecadal frequencies (**Fig. 9**). Running correlations (31-year) reveal a highly non-stationary relationship between Ifaty $\delta^{18}O_{seawater}$ and ENSO, switching between negative and positive correlations (**Fig. S14**).

385        The foregoing analysis suggest that the $\delta^{18}O_{seawater}$ record at Ifaty may be connected with salinity variability upstream in the southwestern Indian Ocean through propagation of ocean Rossby waves, potentially steered by interannual variability climate modes. We thus compared reconstructed Ifaty $\delta^{18}O_{seawater}$ with southwestern Indian Ocean coral-derived $\delta^{18}O_{seawater}$ records, mainly following the South Equatorial Current pathway (**Fig. 11**). The interannual atmospheric and oceanic anomalies intrinsic to each coral reef may differ related to local wind/evaporation and/or rainfall variations. This is confirmed by northern

Madagascar, Mayotte, and La Reunion SODA salinity data, indicating year-to-year differences in mean salinities between sites since 1958 (**Fig. S6**). Interestingly, southwestern Indian Ocean coral-derived $\delta^{18}O_{seawater}$ records largely covary with local rainfall or rainfall over adjacent land areas (**Fig. S15**). Not surprisingly, tracking $\delta^{18}O_{seawater}$ records across thousands of kilometers shows varying leads or lags, preventing us from establishing robust statistical relationships (**Figs. 11a-d; Fig. S16-**

S17). All western Indian Ocean $\delta^{18}O_{seawater}$ records show no linear trend towards more or less saline conditions yet are dominated by interannual to multidecadal variability. The closest coral record to Ifaty is derived from Mayotte in the northern Mozambique Channel at a distance of 1180 km (Zinke et al., 2008), dating back to 1882. Remarkably, Mayotte and Ifaty $\delta^{18}O_{seawater}$ largely agree in terms of decadal and long-term variations between 1882 and 1994, with a lower agreement in the early 20$^{th}$ century (**Fig. 11a**). In particular, the low $\delta^{18}O_{seawater}$ in Mayotte in the early 1970s was not pronounced at Ifaty. A composite $\delta^{18}O_{seawater}$ record from Antongil Bay (northeast Madagascar; approximately 2400km from Ifaty) covaried with the Mayotte data between 1965 and 1994 (r=0.66, p<0.0001), and its decadal variability range overlaps with Ifaty $\delta^{18}O_{seawater}$ (**Fig. 11b**). A record from La Reunion $\delta^{18}O_{seawater}$ directly within the flow path of the SEC (2900 km from Ifaty) also largely overlaps with decadal variability and long-term trends in Ifaty $\delta^{18}O_{seawater}$ changes between 1914 and 1994 (**Fig. 11c**). However, the regime shift in the 1950s in the La Réunion $\delta^{18}O_{seawater}$ record is not as pronounced at Ifaty. Despite the vast spatial differences between all sites, we find overall comparable decadal changes and trends. We computed the absolute differences between Ifaty $\delta^{18}O_{seawater}$ and the records from Mayotte, Antongil Bay and La Réunion taking into account the full uncertainties of individual reconstructions (**Fig. S17**). This analysis reveals that the absolute difference is smaller than the individual uncertainties, thus the $\delta^{18}O_{seawater}$ ranges for all western Indian Ocean sites fully overlap and are indistinguishable. We propose that $\delta^{18}O_{seawater}$ is modified by site-specific atmospheric (P-E) and oceanic variability, and likely involve temporal lags, in agreement with salinity data (**Fig. S6**).

## 4. Discussion

### 4.1 Fidelity of reconstructed SST and $\delta^{18}O_{seawater}$

This study set out to characterise SST and SSS variability in the greater AC region since the Little Ice Age, beyond current observational capabilities, through coral $\delta^{18}O_{seawater}$ and Sr/Ca-SST reconstructions. Although Sr/Ca-SST shows statistically significant correlations with the observational SST data at Ifaty and within the AC core region; that for coral $\delta^{18}O$-SST with observational SST data was overall higher (Tab. S1). Ifaty composite coral $\delta^{18}O$-SST was previously shown to be highly correlated with large-scale SST in the southern Indian and Atlantic Ocean (Zinke et al., 2014). However, note that $\delta^{18}O$-SST is measured on coral $\delta^{18}O$, centered and converted to SST units, *i.e.* it includes both SST and $\delta^{18}O_{seawater}$. The lower correlations of Sr/Ca-SST with observational data may be related to Sr/Ca-SST recording reef-scale SST at individual sites (Ifaty and Tulear reef) while coral $\delta^{18}O$ bears an imprint from larger-scale processes in the region (SST, P-E, ocean advection). This, in turn, may have resulted in overall higher correlations of coral $\delta^{18}O$-SST with instrumental SST. Furthermore, the mean annual Sr/Ca-SST record is largely based on yearly sampled growth increments, with the exception of multidecadal periods previously analysed bimonthly (Zinke et al., 2004). $\delta^{18}O$ in the Ifaty-4 core, which spans the period beyond overlap with Ifaty-1 and Tulear-3 cores, was sampled bimonthly throughout. Thus, the overall higher sample resolution in Ifaty-4 and the regionally

more homogeneous coral $\delta^{18}O$ signals between cores may have improved the overall agreement with instrumental SST. Furthermore, the standard deviations of mean annual SST at Ifaty is only 0.25°C which leads to a lower signal to noise ratio in annual Sr/Ca-SST estimates. With Sr/Ca-SST having an analytical uncertainty of ±0.15°C, the correlation between ERSST and coral Sr/Ca-SST should range between 0.3 and 0.4 following the method of Smerdon et al. (2016), exactly what we obtained in this study (Figs. S1 and S2). However, the mean annual Sr/Ca-SST reconstruction compares favourably with previous results, both of which show lower Sr/Ca-SST in the 19[th] and 18[th] century than $\delta^{18}O$-SST in multidecadal periods with bimonthly Sr/Ca data in core Ifaty-4 (Zinke et al., 2004). Furthermore, the $\delta^{18}O_{seawater}$ reconstruction, based on Sr/Ca-SST and ERSST5 (or HadISST), do not differ substantially between 1870 and 1995 (**Figs. 3 and S5**). We are thus confident that the Sr/Ca-SST provides a robust SST record for the $\delta^{18}O_{seawater}$ reconstruction.

The hindcast (1958-2018) simulation with ocean/sea-ice model configuration INALT20 at 1/20° horizontal resolution (Schwarzkopf et al., 2019) supports our hypothesis that the temporal variability of SST and salinity at Ifaty-Tulear (in simulation and observations) is representative for SST and salinity in the wider AC region on interannual to sub-decadal time scales. However, salinity variability in the greater AC region appears to be highly uncertain from an observational and modelling perspective, with little agreement between salinity products. This may point to uncertainties in atmospheric reanalysis products used in the simulations and/or the scarcity of historical salinity observations in the region that feeds the salinity database (Giese and Ray, 2011). A comparison between SODA (Giese and Ray, 2011) and EN4 (Good et al., 2013) salinity data from the tropical and subtropical Indian Ocean reveals a better agreement in the tropics (**Tab. S3**). The latter may point to higher variability in salinity in the subtropics along the Rossby wave track, including the southern Mozambique Channel and greater AC region, due to more vigorous eddy activity and strong air-sea interactions paired with horizontal and vertical advection (Schott et al., 2009). However, a comprehensive assessment of the strength and weaknesses of salinity products for the southwestern Indian Ocean is beyond the scope of this paper. The model simulation further confirms our coral $\delta^{18}O_{seawater}$ reconstruction findings; namely that there is no clear, direct relationship between salinity and net surface heat/freshwater fluxes at Ifaty-Tulear and the AC region (**Fig. 5**). Coral $\delta^{18}O_{seawater}$ and salinity implies that upstream rainfall in the northern Mozambique Channel and east of Madagascar is introducing negative salinity anomalies which are transported into the southern Mozambique Channel by Rossby waves and eddy transport (**Fig. 7**). Hence, these findings support the idea that SST and salinity variability at Ifaty-Tulear and in the wider AC region is dominated by oceanic heat and freshwater/salt anomaly propagation by Rossby waves.

## 4.2. **Large-scale drivers of AC core region SST and salinity**

In this study, we showed that reconstructed $\delta^{18}O_{seawater}$, after applying a 5-year low-pass filter, agree with the results from salinity observations and reveal a positive correlation (detrended: r= 0.67, p= 0.0063) with the southern Indian Ocean (10-40°S, 50-100°E) ICOADS zonal wind stress. This relationship implies that negative anomalies in zonal wind stress translate into a strengthening of easterly anomalies across the southern Indian Ocean along the South Equatorial Current (SEC)

route driven by the trade winds, thus resulting in lowered salinity levels (lowered $\delta^{18}O_{seawater}$). The 1-year lagged correlation between $\delta^{18}O_{seawater}$ and Indian Ocean P-E anomalies indicates a negative correlation with the southern Indian Ocean (Mascarene Islands) and northern MC (**Fig. 7**). This relationship implies that low salinity anomalies in the Ifaty-Tulear and AC region may be derived from ocean advection by the South Equatorial Current. This result is in agreement with studies based on instrumental data and model simulations for AC transport and SST (Schouten et al., 2002; de Ruijter et al., 2005; Backeberg et al., 2010; Biastoch et al., 2009; Rouault et al., 2009; Loveday et al., 2014; Beal & Elipot, 2016). These studies found that large-scale variability upstream of the AC is related to southern Indian Ocean wind stress curl in the trade wind belt (Backeberg et al., 2012). Our analysis of lagged correlations with the Multivariate ENSO index reveals a twelve to eighteen months lag of $\delta^{18}O_{seawater}$ and salinity at Ifaty and the AC region. These results agree with the 2-year lag found for the Agulhas leakage region, which is further downstream from our coral sites and implies a role for ENSO in interannual timescales of ocean connectivity (Potemra, 2001; Schouten et al., 2002; Wijfels & Meyers, 2004; de Ruijter et al., 2005; Palastanga et al., 2006; Putrahasan et al., 2016; Paris et al., 2018). Salinity studies in the Agulhas leakage region have revealed a 20 to 26 months lag to ENSO forcing, whereby a fresh anomaly is replaced two years later by a saline anomaly following El Niño, and vice versa for La Niña (Paris et al., 2018; Trott et al., 2021). The lagged relationship between ENSO and $\delta^{18}O_{seawater}$ is robust up to 1750 based on analysis with two palaeo-ENSO reconstructions. In summary, $\delta^{18}O_{seawater}$, together with observational data and ocean model simulations, imply a lagged response to large-scale wind forcing in response to southern Indian Ocean trade wind changes. However, ENSO alone only explains a small fraction of the AC region SST and salinity variability (Elipot and Beal, 2018). Extratropical atmospheric and local stochastic variability are also likely important factors affecting such SST and salinity fluctuations (Putrasahan et al., 2016). According to Elipot & Beal (2018), ENSO explains 11.5% of AC transport and 20-30% of sea surface height variability, while other southern hemisphere atmospheric modes explain 29% of such variance. This highlights the complex atmospheric and oceanic dynamics within the AC region and the southern Indian Ocean.

Having established that wind forcing is a likely candidate for driving large-scale ocean advection in the AC region, we now discuss the trends and variability in coral-derived SST and $\delta^{18}O_{seawater}$ reconstructions. The new $\delta^{18}O_{seawater}$ reconstruction is dominated by multidecadal to centennial variability throughout the 334-year record, punctuated by strong interannual and interdecadal variability. Similar to Sr/Ca-SST, $\delta^{18}O_{seawater}$ shows largest interannual variability during the Late Maunder Minimum (LMM; 1670-1715) when solar activity was low, which is in agreement with previous studies based on Ifaty-4 coral $\delta^{18}O$ (Zinke et al., 2004, 2014). Multitaper spectral analysis and moving 30-year standard deviations for both reconstructed mean annual Sr/Ca-SST and $\delta^{18}O_{seawater}$ confirm higher interannual variability during the so-called Little Ice Age (1661-1900) than 20th century (**Figs. 8 and 9**). Nevertheless, part of the higher interannual variability between 1661 and 1900 may stem from the use of a single coral core and overall higher magnitude of variability in Sr/Ca-SST compared to $\delta^{18}O$-SST (**Fig. 2**). Spectral coherence analysis suggests that Ifaty-Tulear Sr/Ca-SST and $\delta^{18}O_{seawater}$ is coherent with palaeo-ENSO reconstructions at frequencies between 2 and 4 years, as well as decadal bands ranging between 8-30 years (**Fig. 9; Fig. S12-S13**). However, the band-pass filtered $\delta^{18}O_{seawater}$ reconstruction and palaeo-ENSO reconstructions for interannual to multi-

decadal periodicities showed various levels of agreement and disagreement with individual ENSO reconstructions. Wavelet coherence analysis indicated a high level of agreement in the 20th century in the 8-16 years frequency band yet with a phase lag between 90 and 180 degrees. The 8-16 years frequency is present in all ENSO records and both Ifaty-Tulear $\delta^{18}O_{seawater}$ and Sr/Ca-SST, thus it appears to be an important timescale of variability in the greater Agulhas region. The latter was also identified in instrumental SST records and coral-derived SST reconstructions at Ifaty (Zinke et al., 2004, 2009, 2014). Running correlations (31-year) revealed a highly non-stationary relationship between Ifaty $\delta^{18}O_{seawater}$ and ENSO, switching between negative and positive correlations (**Fig. S14**). A non-stationary ENSO relationship was also found by Zinke et al. (2004) for the high-resolution Ifaty-4 $\delta^{18}O$ record between 1870 and 1995. The latter was attributed to a non-stationary relationship between Ifaty SST with ENSO.

Indian Ocean coral records from the southeast and southwest also indicate a period of larger interannual and decadal variability around the turn of the 17th century (Damassa et al., 2006; Abram et al., 2020; Leupold et al., 2020). The larger interannual variability is ascribed to higher ENSO and IOD variability at this time, as confirmed by the eastern and western Indian Ocean (Damassa et al., 2006; Abram et al., 2020; Leupold et al., 2020) and central Pacific (Cobb et al., 2013) coral reconstructions. A (bi)monthly coral $\delta^{18}O$ record from Ifaty indicated a stronger relationship with ENSO when ENSO variability was high during the observational period (Zinke et al., 2004). The dominant interannual frequency band in coral $\delta^{18}O$ was 3.9 years, which is typical for ENSO variability. The bandpass filtered (4 years) bimonthly coral $\delta^{18}O$ record revealed substantial amplitudinal variations between 1680-1720, 1760-90, 1870-1920, 1930-40, and 1960-1995, which were ascribed to ENSO (Zinke et al., 2004). The larger interannual swings in mean annual reconstructed Ifaty Sr/Ca-SST and $\delta^{18}O_{seawater}$ presented here may therefore be partly ascribed to ENSO. ENSO excites oceanic Rossby waves, leading to warming during El Niño and cooling with La Niña in the southwestern Indian Ocean (East and North of Madagascar), usually one season after ENSO has peaked (Schott et al., 2009). The observational SST record has also revealed a 1- to 2-year lag between ENSO and AC and AL SST, respectively (Putrasahan et al., 2016). Salinity in the AC region also apparently lags El Niño and La Niña events for up to 2 years (Trott et al., 2021). The ENSO-related SST signal develops faster than salinity, most probably related to ENSO's influence on atmospheric processes in the southern Indian Ocean (Putrasahan et al., 2016). The mean annual $\delta^{18}O_{seawater}$ record agrees with these previous findings of a twelve to eighteen months lag with the Multivariate and palaeo-ENSO indices. In summary, these findings raise the possibility that interannual $\delta^{18}O_{seawater}$ variability may be steered at least partly by ENSO.

Decadal variability in the southwestern Indian Ocean SST is said to be related to ENSO-like decadal variability driven by sea-level pressure and wind fields (Reason, 2001; Schott et al., 2009). To this end, coral $\delta^{18}O$ records from Ifaty-Tulear and La Réunion indicate strong spatial and temporal covariance with ENSO-like decadal variability and the PDO in SST/SLP fields (Crueger et al. 2009). Furthermore, multitaper spectral analysis indicates a 57-year (42 to 68-year band) multidecadal frequency in Ifaty $\delta^{18}O_{seawater}$ and a 33-year (27 to 42-year band) frequency in Sr/Ca-SST (**Fig. 8**). Both multidecadal variations indicate higher amplitudes during the Little Ice Age compared to the 20th century (**Fig. 8**). Furthermore, the decadal to

interdecadal variability for Ifaty-Tulear $\delta^{18}O_{seawater}$ and Sr/Ca-SST is coherent with palaeo-ENSO indices in the 8-16 years frequency band (partly also the 30 years band), as shown above, and indicates higher power between 1650 and 1850 (**Fig. 9; Figs. S12-S13**). A coral Sr/Ca-SST record from Rodrigues island in the Mascarene Island chain and a coral luminescence record from Antongil Bay (northeast Madagascar) also demonstrate a link with the PDO to Indian Ocean SST and rainfall, respectively (Grove et al., 2013; Zinke et al., 2016). Similarly, Damassa et al. (2006) found enhanced multidecadal variability in a 17[th] century coral $\delta^{18}O$ record from Mafia Island (Tanzania) partly ascribed to ENSO-like decadal variability. A coral $\delta^{18}O_{seawater}$ reconstruction from Mayotte (northern MC) is characterized by interdecadal variability in the 18 to 25 years band, while the Ifaty-4 coral $\delta^{18}O$ record has such variability centered around 16-18 years (Zinke et al., 2004, 2008, 2009). While the La Reunion coral $\delta^{18}O$ indicates coherence with ENSO-like decadal variability, the synthesis of tropical and subtropical western Indian Ocean coral $\delta^{18}O$ records also show pronounced decadal variability, but not necessarily connected to ENSO (Pfeiffer et al., 2004; Zinke et al., 2009). Thus, similar to ENSO, Pacific decadal variability appears to steer decadal variability in the wider southwestern Indian Ocean and AC core region, as observed in instrumental data (Reason, 2001, 2002; Schott et al., 2009). However, regional modes of variability acting on interannual to decadal time scales in the subtropical and mid-latitude Indian Ocean may interact with ENSO-like decadal variability and mask clear relationships (Elipot & Beal, 2018).

We consider published $\delta^{18}O_{seawater}$ reconstructions from La Reunion (Mascarene Islands; Pfeiffer et al., 2004, 2019), Antongil Bay (northeast Madagascar; Grove et al., 2012), and Mayotte (Comoro Islands; Zinke et al., 2008), obtained along the South Equatorial Current pathway in the southern Indian Ocean, to assess whether these records share variability with our new Ifaty $\delta^{18}O_{seawater}$ reconstruction (**Fig. 11; Figs. S16-S17**). Considering the uncertainties in $\delta^{18}O_{seawater}$ reconstructions, we find overall comparable trends across the southern Indian Ocean locations and regional nuances. This implies that regional processes (*e.g.* P-E balance, horizontal and vertical advection, ocean eddies) strongly modify surface salinity and therefore $\delta^{18}O_{seawater}$ while being advected across the southern Indian Ocean. SODA salinity data confirm the latter (Fig. S6). For instance, at Mayotte, $\delta^{18}O_{seawater}$ is strongly related to local rainfall station data between 1950 and 1995 (Zinke et al., 2008). The Mayotte coral record was obtained from a relatively shallow lagoonal site which is probably more sensitive to local rainfall or the P-E balance affecting $\delta^{18}O_{seawater}$ (**Fig. S15**). The Antongil bay site in northeast Madagascar is under the influence of several smaller river catchments that drain into the enclosed bay area (Grove et al., 2012). Thus, salinity and $\delta^{18}O_{seawater}$ may be strongly affected by both regional rainfall and river freshwater discharge (**Fig. S15**). The La Réunion coral record was obtained from the leeward side of the island where the flow of the South Equatorial Current is diverted around the island (Pfeiffer et al., 2004, 2019). The high topographic relief of La Réunion (>3000m) blocks rainfall derived from east-southeast from reaching the leeward island. Thus, salinity and therefore $\delta^{18}O_{seawater}$ mid-20th century freshening may not be a prime South Equatorial Current signature. Local freshwater discharge by rivers or groundwater from the high mountainous terrain of La Réunion may also contribute to local $\delta^{18}O_{seawater}$ variability. However, La Réunion $\delta^{18}O_{seawater}$ shows a negative correlation with rainfall on the eastern coast of Madagascar, both regions being under the influence of the trade winds transporting moisture to Madagascar (**Fig. S15**). Nevertheless, the regional comparison between $\delta^{18}O_{seawater}$ records suggests that such reconstructions

bear huge potential to unlock past interannual and decadal changes in regional surface ocean hydrology and ocean transport dynamics across the southern Indian Ocean beyond the short instrumental record.

**5. Conclusions**

The aim of this study was to unravel SSS variability in the AC region since the Little Ice Age; this based on paired coral Sr/Ca and $\delta^{18}$O records obtained from the Ifaty-Tulear reef southwest of Madagascar. Our new 334 year (1661-1995) annual $\delta^{18}O_{seawater}$ composite record from Ifafy-Tulear traces surface salinity of the southern MC and AC core region from SODA since 1958. A high-resolution ocean model confirms that our study site is optimally located to trace AC region SSS and SST

variability. Although the interannual changes in ocean model salinity only partly agree with observations and coral $\delta^{18}O_{seawater}$, it demonstrates the huge potential of combining high resolution ocean model studies with paleoclimate reconstructions to improve upon the mechanistic understanding of ocean dynamics at the scale of coral reefs and beyond. Ifaty-Tulear $\delta^{18}O_{seawater}$ and AC salinity appear not to be driven by regional precipitation-evaporation changes, but rather by upstream changes. We show that $\delta^{18}O_{seawater}$ variability is likely driven by changes in the large-scale wind forcing (zonal wind stress) in the southern

Indian Ocean on interannual to decadal time scales, as has been suggested based on short observational studies. The $\delta^{18}O_{seawater}$ at Ifaty co-varies with the southwestern Indian Ocean coral-derived $\delta^{18}O_{seawater}$ records along the path of the South Equatorial Current, suggesting that ocean advection may significantly contribute to salinity changes in the wider southwestern Indian Ocean. Ocean advection may be assisted by the wind stress changes along the South Equatorial Current pathway in the southwestern Indian Ocean, modulated by ENSO or IOD. Both $\delta^{18}O_{seawater}$ and SST at Ifaty show characteristic interannual

variability of between 2 to 4 years, most likely driven by ENSO. ENSO changes are shown to lead Ifaty and AC salinity and $\delta^{18}O_{seawater}$ by twelve to eighteen months, in agreement with previous studies on Agulhas leakage salinity and SST. The $\delta^{18}O_{seawater}$ at Ifaty and SST reconstructions reveal the highest interannual variability during the Little Ice Age, between 1700 and 1900, highest during the 1670-1710 period. Interdecadal variability is also coherent between $\delta^{18}O_{seawater}$, Sr/Ca-SST and ENSO reconstructions and was also enhanced during the 1670-1710 period. Other Indo-Pacific coral studies have also

indicated a high variability in Indian Ocean Dipole and ENSO variability at the turn of the 16[th] to 17[th] century. Our study demonstrates that surface ocean salinity, derived from coral $\delta^{18}O_{seawater}$, in the AC region underwent strong interannual and decadal changes since 1661, testifying the highly dynamic nature of this particular oceanic region. The interdecadal 8-16 years frequency band appears as a prominent time scale of variability in the greater Agulhas region surface ocean with strong links to ENSO-like decadal variability. A wider network of coral $\delta^{18}O_{seawater}$ and SST reconstructions spanning the AC region and

the Mozambique Channel would be instrumental in ground-truthing the role of ocean advection in driving surface salinity and SST variations in the southern Indian Ocean. Ultimately, such long records may help assess the variability of Agulhas leakage salinity and SST over multidecadal to centennial timescales, as well as the resulting potential modulations of the AMOC. Thus, developing paired coral Sr/Ca sand $\delta^{18}$O time series for such key regions of ocean dynamics may improve our understanding of extreme events, their impacts on ecosystems and societies, and their drivers.


**6. Acknowledgements**

A Royal Society Wolfson Fellowship, grant RSWF-FT-180000, and an Honorary Fellowship at the University of Witwatersrand supported J.Z. This work was supported as part of the SINDOCOM grant under the Dutch NWO program 'Climate Variability', grant 854.00034/035. Additional support comes from the NWO ALW project CLIMATCH, grant

820.01.009, and the Western Indian Ocean Marine Science Association through the Marine Science for Management programme under grant MASMA/CC/2010/02 led by Professor Chris Reason and Jens Zinke. We thank the VU University Amsterdam (Netherlands) for assistance with stable isotope analysis, especially Suzan Verdegaal. With thank Wolf-Christian Dullo and Georg Heiss from GEOMAR Helmholtz Centre for Ocean Research Kiel and the Free University of Berlin, respectively, and the EU TESTREEF party for sampling the coral cores in 1995.


**7. Data availability**

Data generated for this publication are provided as supplementary tables 3 and 4 with the Supplementary Information. Data will be stored publicy with https://www.ncei.noaa.gov/products/paleoclimatology once the article has been accepted.

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

**Table 1:** Linear least squares correlation between annual mean salinity from SODA 2.1.6 and Ifaty-Tulear $\delta^{18}O_{seawater}$ (from coral Sr/Ca and $\delta^{18}O$) between 1958 and 1995 (March to February) for detrended and non-detrended data (p-values in brackets; CI= 95% confidence interval, N= number of observations; DoF= degrees of freedom, taking into account autocorrelation in SODA salinity data).

| | IFA SODA SSS (detrended) | IFA SODA SSS (not detrended) | AC SODA SSS (detrended) | AC SODA SSS (not detrended) |
|---|---|---|---|---|
| $\delta^{18}O_{seawater}$ | 0.50 (0.008) | 0.63 (0.001) | 0.57 (0.002) | 0.70 (<0.001) |
| **95% CI** | 0.21 to 0.62 | 0.41 to 0.71 | 0.26 to 0.74 | 0.50 to 0.77 |
| **N** | 36 | 36 | 36 | 36 |
| **DoF** | 18 | 18 | 18 | 18 |


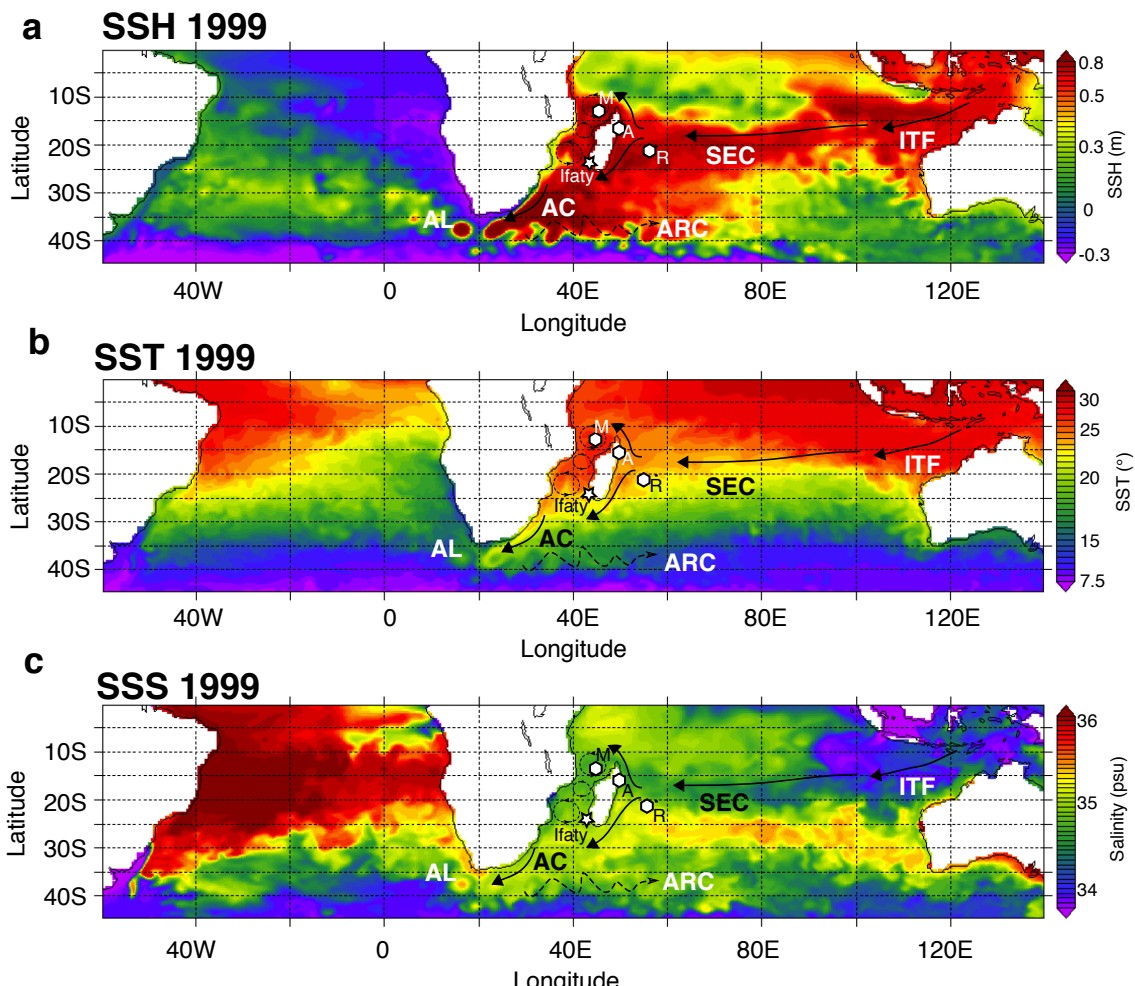

**Figure 1 – Surface ocean connectivity between southwest Madagascar and the Agulhas Current.** a) Sea surface height (SSH), b) temperature (SST) and c) salinity (SSS) across the southern Indian Ocean and Atlantic from SODA reanalysis for August 1999 (Giese and Ray, 2011). Star marks the location of our study location at Ifaty and Tulear coral reefs. White filled polygons indicate location of coral records used in comparison to the Ifaty-Tulear reconstruction: M= Mayotte, A= Antongil Bay and R= La Réunion. AC= Agulhas Current, AL= Agulhas Leakage, ARC= Agulhas Return Current, SEC= South Equatorial Current, ITF= Indonesian Throughflow. Mozambique Channel mesoscale eddies indicated.

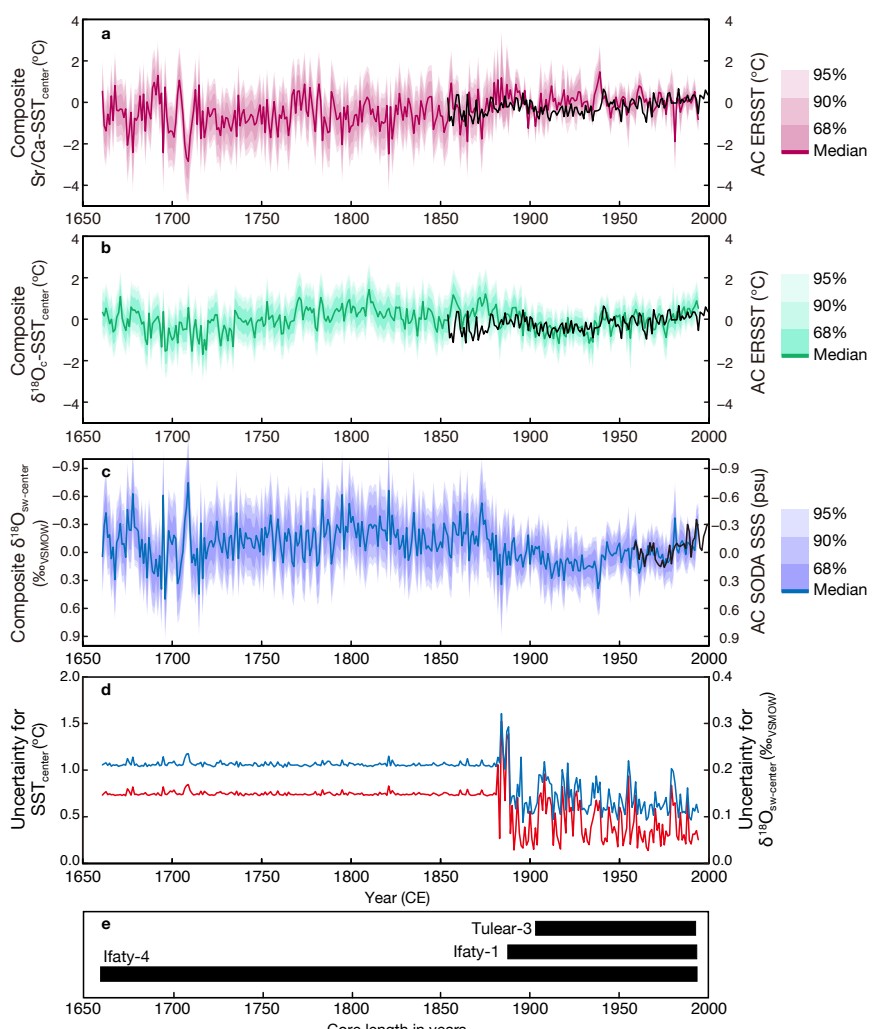

**Figure 2 - Sea surface temperature and δ¹⁸O$_{seawater}$ reconstruction for the greater Agulhas Current region.** a) Coral composite annual SST anomaly reconstruction (red) for southwestern Madagascar (red shading shows 68 to 95% confidence intervals) based on Sr/Ca (red) compared to Agulhas Current (AC) core region data from ERSSTv5 (black), b) same as a) yet for coral composite δ¹⁸O-SST anomaly reconstruction (blue with shading showing 68 to 95% confidence intervals), c) Coral composite δ¹⁸O$_{seawater}$ anomaly reconstruction (blue with shading showing 68 to 95% confidence intervals) for southwestern Madagascar compared to Agulhas Current core region SODA reanalysis salinity data (black). d) Uncertainties in reconstructed Sr/Ca-SST (centered; red) and δ¹⁸O$_{seawater}$ (centered; blue) based on Monte Carlo simulation, e) Time coverage of individual coral core records. All anomalies computed relative to the 1961 to 1990 period.

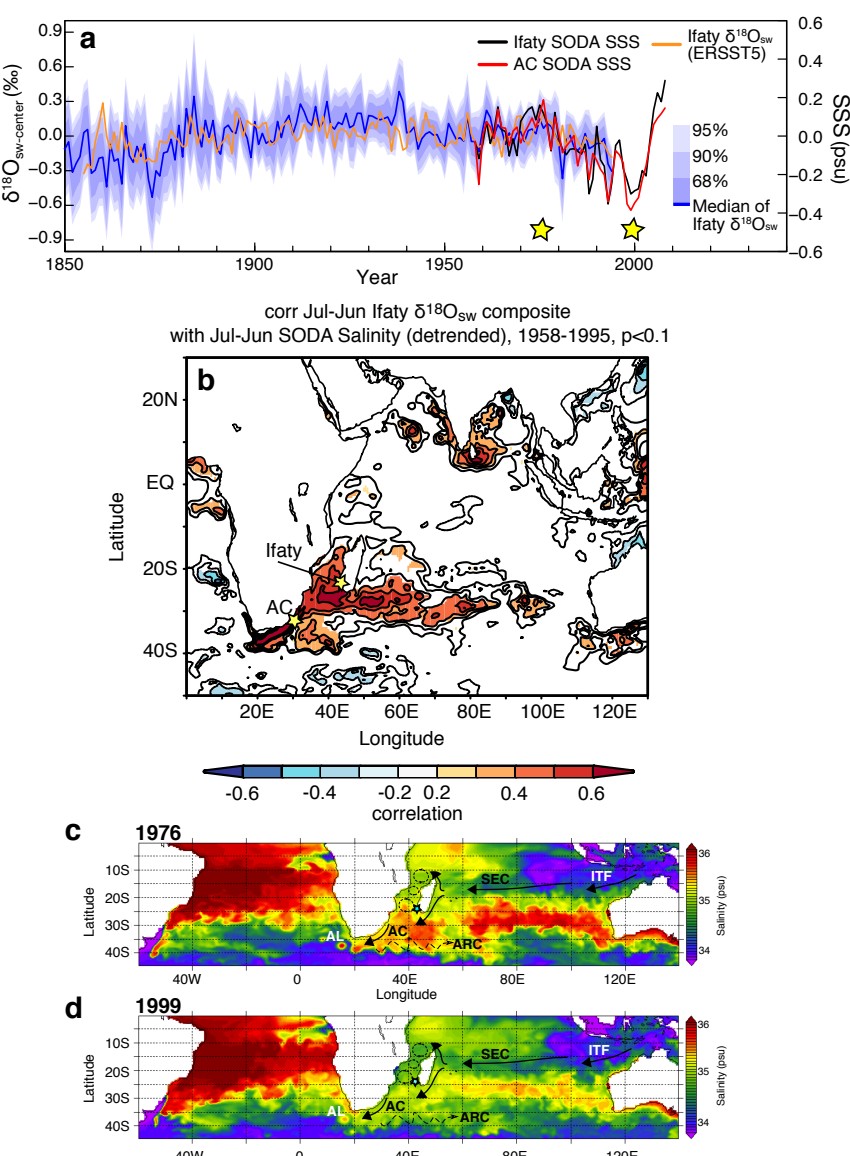

**Figure 3 – Sea surface salinity data from SODA reanalysis in the greater Agulhas Current and the Ifaty-Tulear region.** a) Time series of salinity (March to February) for the grid box surrounding Ifaty and Tulear reef (black), the Agulhas Current core region (red) and the reconstructed $\delta^{18}O_{seawater}$ from Ifaty and Tulear corals (blue; orange= $\delta^{18}O_{seawater}$ using ERSSTv5 instead of Sr/Ca-SST) with blue shading indicating the 68 to 95% confidence intervals. The yellow star marks a high salinity period in August 1976 and a low salinity period in August 1999 in the greater Agulhas Current (AC) region depicted in panels c and d. b) Spatial correlation of deterned Ifaty-Tulear $\delta^{18}O_{seawater}$ with SODA salinity. c and d) Blue filled star marks the location of the coral core site. Note the accumulation of high saline anomalies in the southwestern Indian Ocean off the southeast coast of South Africa in 1976 (c) and vice versa for 1999 (d).

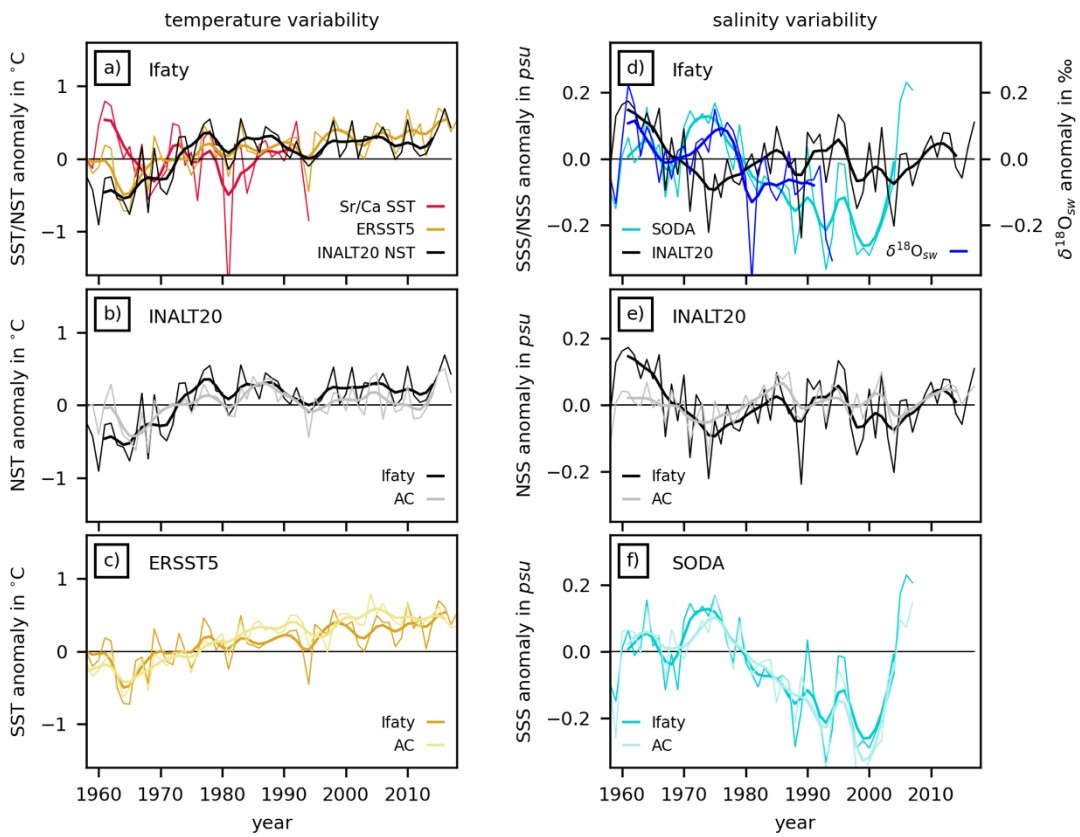

**Figure 4 – Reconstructed and simulated co-variability of temperature and salinity in the Ifaty-Tulear and AC core regions**. a) SST at Ifaty reconstructed from coral Sr/Ca (red), b) simulated with INALT20 (black), and c) obtained from ERSST5 (dark yellow), as well as in b) NST in AC core region simulated with INALT20 (grey) and obtained from ERSST5 (light yellow); d) SSS at Ifaty reconstructed from coral d18O$_{sw}$ (blue), e) simulated with INALT20 (black), and f) obtained from SODA (dark cyan), as well as in e) NSS in AC core region simulated with INALT20 (grey), and obtained from SODA (light cyan). Shown are annual mean (thin lines) and sub-decadally filtered (7-year Hamming filter) anomalies (referenced to 1961-1990 mean), whereby annual means in ocean model and instrumental data are calculated as March to February averages for better comparison with the coral record.

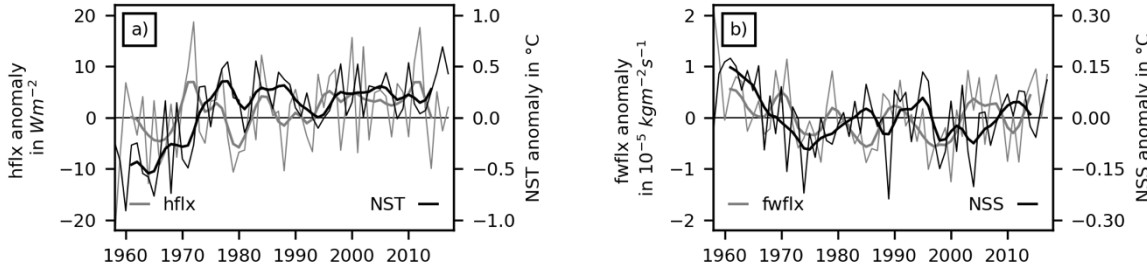

**Figure 5 – Simulated annual mean (thin lines) and sub-decadally filtered (7-year Hamming filter) NST/NSS and surface flux anomalies (referenced to 1961-1990 mean, Figure 7) at IFA.** (a) NST (schwarz) and surface heat flux (hflx, grey, positive downward). (b) NSS (black) and surface freshwater flux (fwflx, grey, positive upward). Annual means are calculated as March to February averages for better comparison with the coral record.

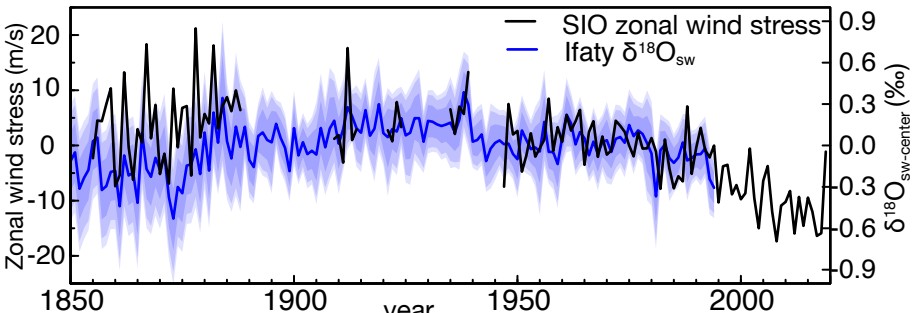

**Figure 6 – Large-scale atmospheric forcing across the southern Indian Ocean trade wind belt driving zonal advection of ocean circulation.** Zonal wind stress averaged over the southern Indian Ocean (10-40°S, 50-100°E) from ICOADS (black) compared to our coral composite $\delta^{18}O_{seawater}$ anomaly reconstruction (blue with shading for confidence intervals) between 1855 and the present. Note the trend towards more easterly wind stress between 1947 and 2008 largely mirrored by our $\delta^{18}O_{seawater}$ reconstruction.

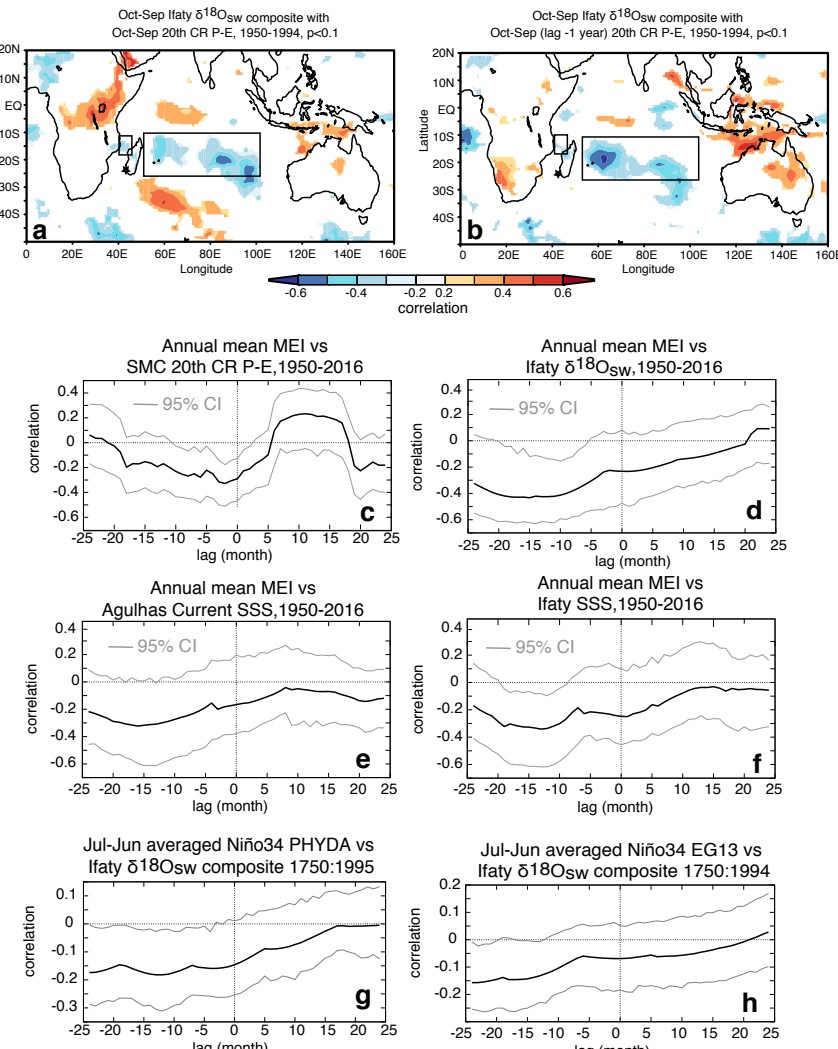

**Figure 7 – Large-scale precipitation-evaporation balance across the southern Indian Ocean.** Spatial correlation, computed with KNMI climate explorer (Trouet and Oldenborgh, 2013), of mean annual Ifaty-Tulear coral composite $\delta^{18}O_{seawater}$ with a) P-E from 20th century reanalysis (Giese et al., 2016) and b) same as a, yet with 12 months lag. Rectangular boxes in panels a and b indicate regions with negative correlations. Only correlations significant at 90% level are coloured. Lagged correlations between the Multivariate ENSO index (MEI) with c) P-E from 20th century reanalysis averaged over 20-25°S, 41-44°E, d) Ifaty coral composite $\delta^{18}O_{seawater}$, e) Agulhas Current (32°S, 32°E) salinity from SODA reanalysis (Giese and Ray, 2011) and f) Ifaty salinity from SODA reanalysis (Giese and Ray, 2011). Note the six to twelve months lag between the MEI and regional hydrology. Negative lag indicates that MEI is leading. 95% confidence intervals indicated (grey lines in c to f). g-h) Lagged correlation between annual mean Ifaty-Tulear $\delta^{18}O_{seawater}$ composite with Nino3.4 index between 1750 and 1995 from g) Steiger et al. (2018; PHYDA) and h) Emile-Geay et al. (2013; EG13). For this period, correlations are significant at 90% level or higher. Beyond 1750, the 15-24-month lagged correlation is no longer significant.

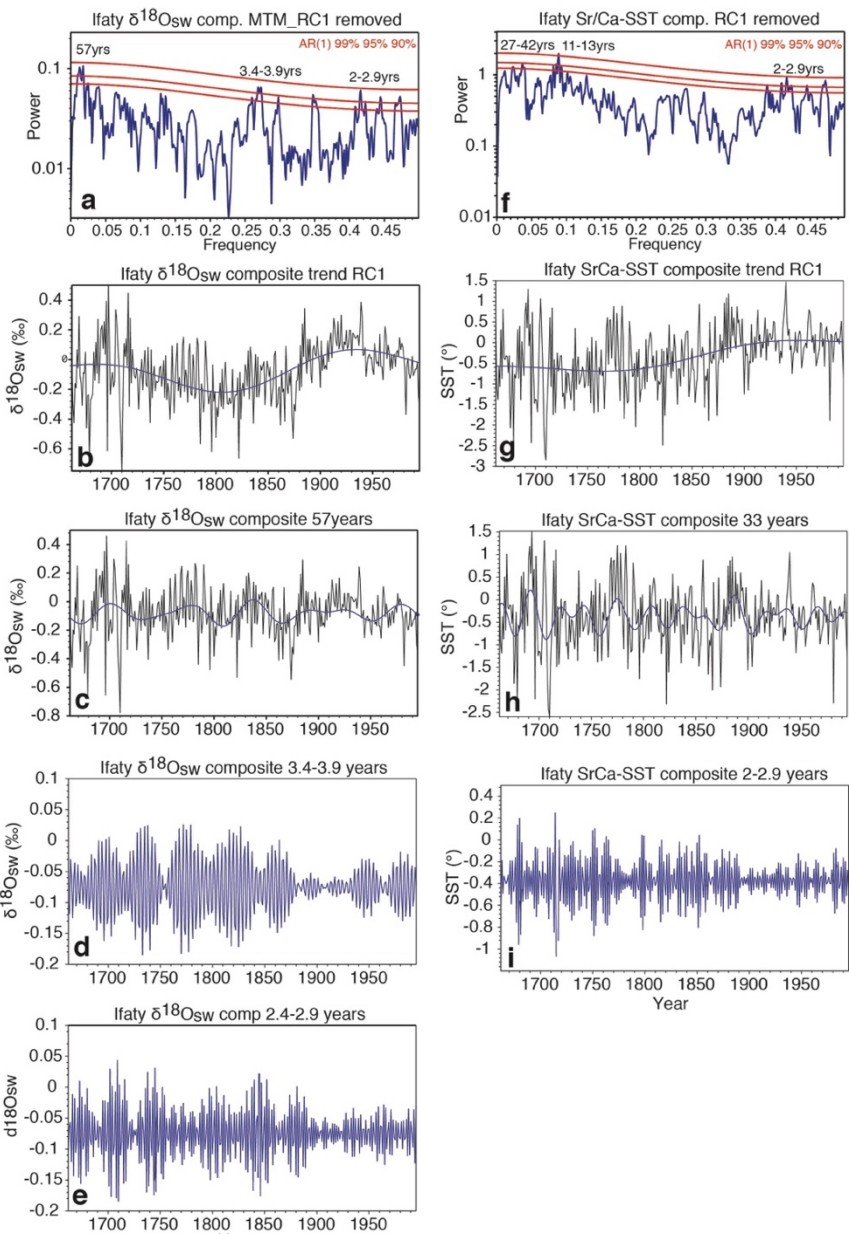

**Figure 8** - Multitaper method spectral analysis (MTM; for detrended data; Torrence and Compo, 1998) and reconstructed components (RCs) of a-e) reconstructed Ifaty $\delta^{18}O_{seawater}$ composite and f-i) Ifaty Sr/Ca-SST composite. b and g) illustrate the long-term trends, c and h) the multidecadal frequencies and d, e, and i) the interannual frequencies. Significance levels for MTM spectra are indicated in a and f.

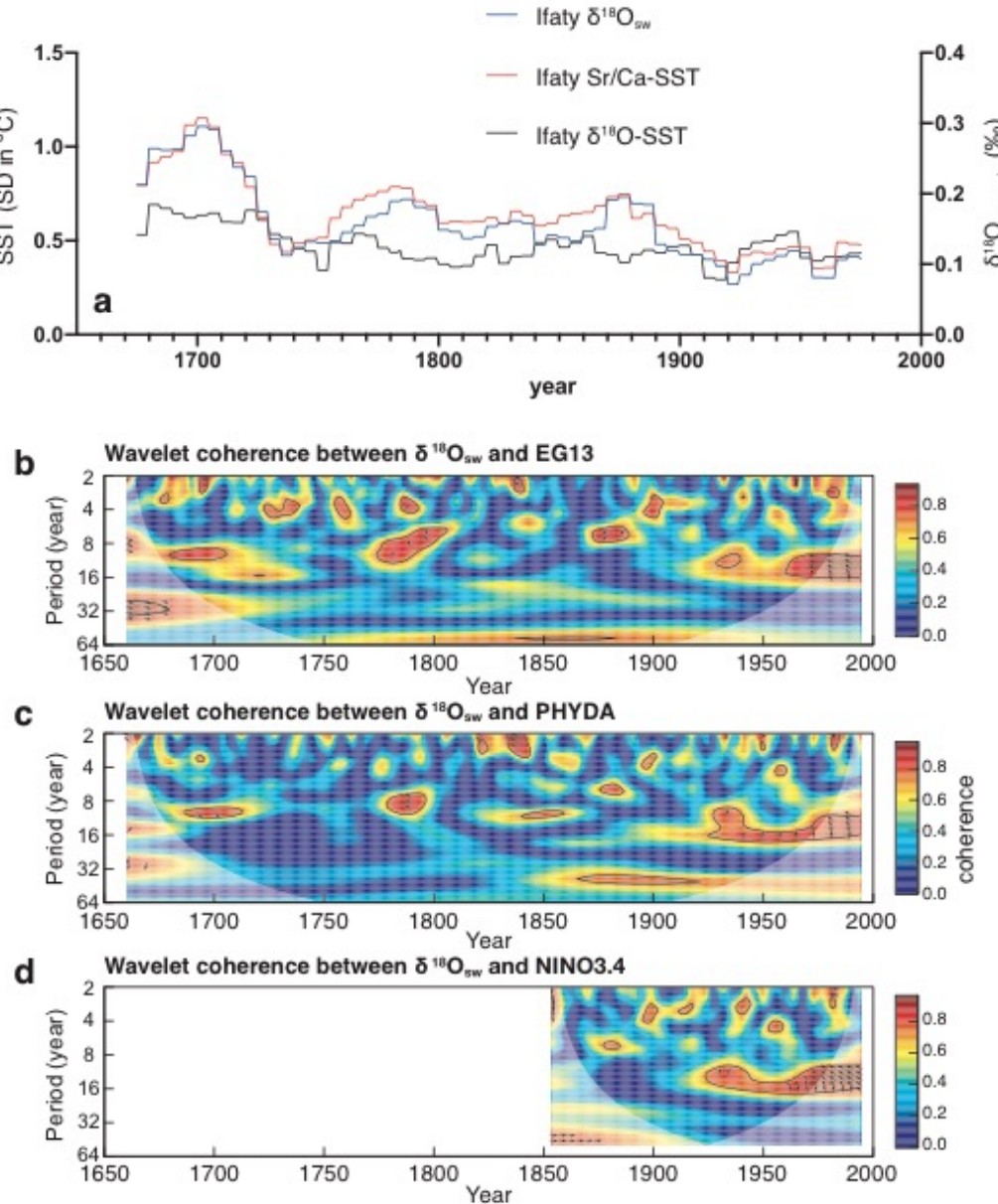

**Figure 9** – a) Moving 30-year standard deviations, stepped by 5 years, of reconstructed Ifaty $\delta^{18}O_{seawater}$ composite (blue), Ifaty Sr/Ca-SST composite (red) and Ifaty $\delta^{18}O$-SST composite (black). b-d) Cross-wavelet coherence (Torrence and Compo, 1998) between Ifaty-Tulear $\delta^{18}O_{seawater}$ composite with Niño3.4 reconstructions of b) Emile-Geay et al. (2013; EG13), c) Steiger et al. (2018; PHYDA) and d) observed Niño3.4 index based on ERSSTv5 between 1661 and 1995. Coherent frequency bands above 95% significance level encircled by black lines and filled red. Phase of coherence indicated by arrows.

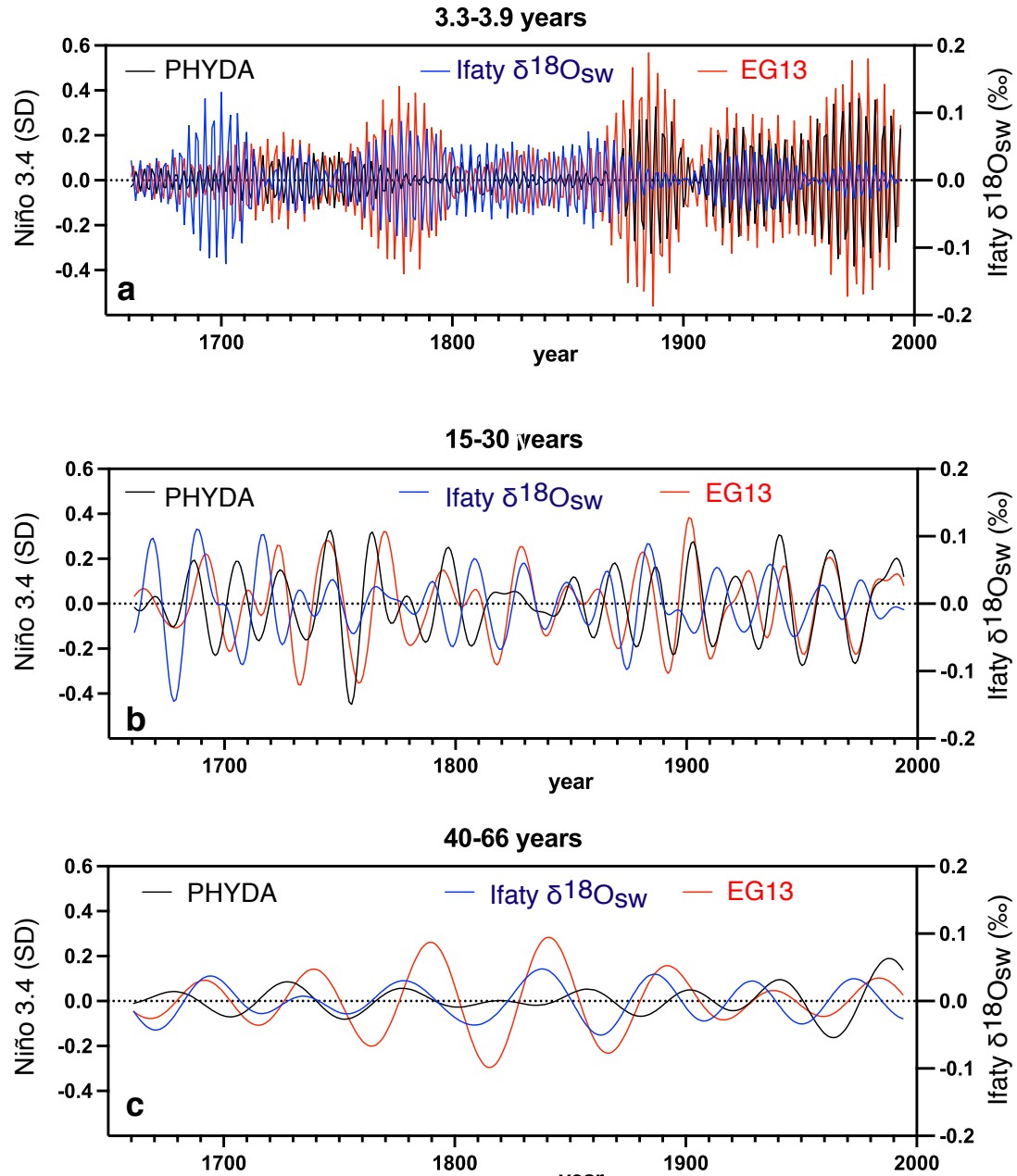

870

**Figure 10** - Bandpass filtered data for Ifaty-Tulear δ18O$_{seawater}$ (blue), Nino3.4 index from Emile-Geay et al. (2013; EG13; red) and Nino3.4 index from Steiger et al. (2018; black; PHYDA) for a) interannual (3.3 to 3.9 years), b) interdecadal (15-30 years) and c) multidecadal (40-66 years) frequency bands.

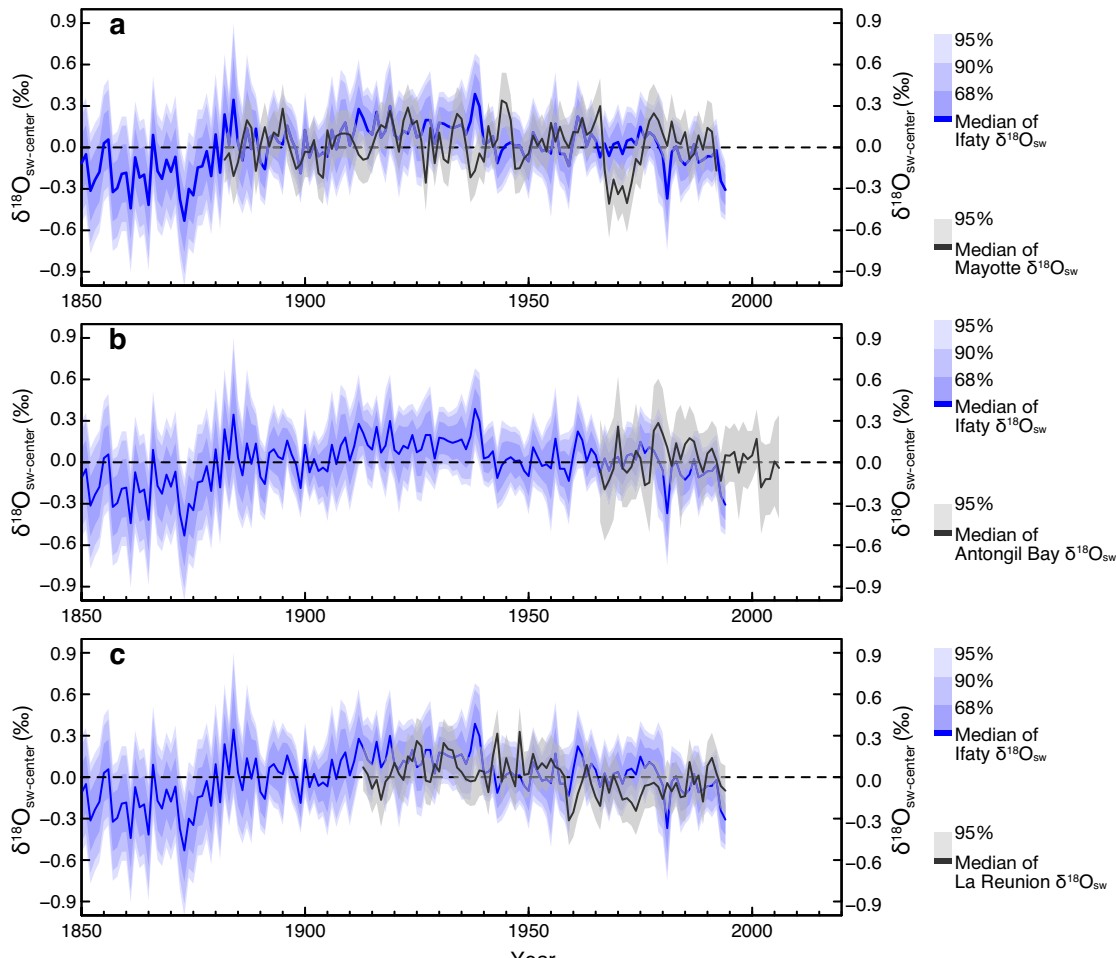

**Figure 11** - Comparison of Ifaty $\delta^{18}O_{seawater}$ (blue) with 68 to 95% confidence interval (light blue) with western Indian Ocean $\delta^{18}O_{seawater}$

880 reconstructions. a) with reconstructed $\delta^{18}O_{seawater}$ for Mayotte (Comoro Archipelago; black; Zinke et al., 2008), b) with reconstructed $\delta^{18}O_{seawater}$ for Antongil Bay (northeast Madagascar; black; Grove et al., 2012), c) with reconstructed $\delta^{18}O_{seawater}$ for La Réunion (black; Pfeiffer et al., 2019). Blue (grey) shading indicates the 68 to 95% confidence intervals of Ifaty (Mayotte, Antongil Bay and La Réunion, respectively) $\delta^{18}O_{seawater}$. Only the correlation between Mayotte and Antongil Bay reconstructed $\delta^{18}O_{seawater}$ is statistically significant.

885