# Peer review of "334-year coral record of surface temperature and salinity variability"

_Climate of the Past, 2021_

## Author Comment (AC1)

**Point-by-point response**
We thank both reviewers for their constructive comments which we address briefly in this
response.
**Reviewer 1**
**1) …why we used annual resolution Sr/Ca record and how to explain relatively low**
**correlation with SST…**
Answer: We opted for multi-core records in this study at annual resolution complemented by
short decadal periods of bimonthly data (published previously in Zinke et al., 2004) in order to
build a long record of $\delta^{18}O_{seawater}$. Our main focus in this study is to assess the interannual to
decadal salinity changes of the greater Agulhas region which have so far not been possible due
to the lack of coupled Sr/Ca and $\delta^{18}O$ data. Unfortunately, a higher resolution Sr/Ca analysis
is not possible at this stage.
Furthermore, instrumental SST is extremely sparse in the region and will by definition not
represent SST at the reef site very well (see Figure 1 of this response). There is no data close
to the reef site for many decades pre-dating the 1970's. The low correlation of Sr/Ca-SST with
ERSST or HadISST may therefore imply that instrumental data coverage precludes us from
making a better judgement. We have illustrated the HadSST4 dataset which has not been
infilled as other SST data products to highlight the number of observations and uncertainties.
We believe that Sr/Ca-SST does reflect local SST well, otherwise the $\delta^{18}O_{seawater}$
reconstructions would not agree with SODA salinity. As such, the $\delta^{18}O_{seawater}$ reconstruction
provides independent proof for the quality of the Sr/Ca-SST data as a local SST record.
However, local SST may be less informative to assess large-scale SST changes in the region.
Note that SST adjusts faster to local atmospheric conditions than $\delta^{18}O_{seawater}$ and salinity, and
thus has a stronger signature of local variability. Nevertheless, it is essential to capture local
SST in order to correctly reconstruct local $\delta^{18}O_{seawater}$. Our relatively low, yet significant
correlation in the Sr/Ca-SST with regional SST is reflected in our Monte Carlo error
propagation approach. Therefore, we have taken this low correlation into account and treated
our resulting reconstruction more conservatively.

[Figure]

**Figure 1** – HadSST4 data for the grid box of Ifaty-Tulear (43°E, 23°S). a) SST anomalies, b)
uncertainty of SST and c) number of observations in SST.

**2) Agreement/disagreement between $\delta^{18}O_{seawater}$ based on Sr/Ca-SST vs. using HadISST…**

Answer: We have indicated the highly significant correlations and significance levels for the detrended $\delta^{18}O_{seawater}$ time series with salinity based on Sr/Ca-SST. In fact, the correlations between $\delta^{18}O_{seawater}$ based on HadISST and salinity are lower, yet still significant. Therefore, despite some year to year variability not being exactly matched in $\delta^{18}O_{seawater}$ based on Sr/Ca-SST, the overall agreement with salinity is statistically robust. Furthermore, HadISST is very sparse for the region as is ERSST5. Hence, we cannot assume that instrumental SST reflect SST at the coral site without significant uncertainties (see comments above). We also like to stress that the $\delta^{18}O_{seawater}$ records based on HadISST or ERSST5 falls within the uncertainty range of our Sr/Ca-SST based $\delta^{18}O_{seawater}$ reconstruction (see Figure 2 of this response). This
adds credibility to the Sr/Ca-SST and our $\delta^{18}O_{seawater}$ reconstruction which are truly
independent realisations from the instrumental SST and salinity data. We now show ERSST5
in Figure 3a to be consistent in choice of SST dataset for the main figures.

[Figure]

**Figure 2** – Ifaty-Tulear $\delta^{18}O_{seawater}$ reconstruction using Sr/Ca-SST (blue), ERSST5 instead of
Sr/Ca-SST (orange) compared to SODA salinity for Ifaty (black) and Agulhas Current (AC;
red).
**3) Manuscript could benefit form more detailed description of model results and**
**potentially re-framing of the aims for the model study…**
Answer: We agree with the reviewer that we can reframe the model analysis to better align
with our goals. Our study benefits from the inclusion of the model results for mainly two
reasons. On the one hand it shows that SST and SSS variability at Ifaty is representative for
interannual to decadal variability in the wider AC region. On the other hand it supports the idea
that surface fluxes are not the main driver of the that variability. Currently, in the main text,
this information is kind of hidden in section 3.1 under the headline "Reconstructed SST and
d18O seawater validation with instrumental and ocean model data". This headline is also
misleading, since we do not use the model data for a validation of the reconstructions in a
traditional sense. During our revision we subdivide the respective old section 3.1 into two new
sections 3.1 and 3.2. The new section 3.1 "Validation of reconstructed Sr/Ca-SST and
d18Oseawater at Ifaty" is focusing on a comparison of the coral reconstructions for SST and
SSS variability at Ifaty with available gridded observation-based products and model data in
the Ifaty-Tulear region. It includes a discussion of the discrepancy between the different
products regarding the exact temporal evolution of SST and SSS caused by limited number of
observations and highlights the best agreement of the coral data with ERSST and SODA. To
simplify this discussion, throughout the whole paper we now only analyse variability based on
annual means averaged from March to February (in the first version of the manuscript the
model part was based on standard January to December means). The new section 3.2
"Representativeness of SST and SSS variability at Ifaty for variability in the wider Agulhas
Current region" then focuses on potential co-variability between SST and SSS at Ifaty and
other locations in the wider Agulhas region. Here, independent of the mentioned disagreements
in the exact temporal evolution, all observational products as well as the model agree that
variability at Ifaty is indeed representative for variability in the AC core region (Figure 3). The
fact that co-variability is not only found in observation-based products but also in the simulated
NST and NSS from an ocean model without data assimilation, supports the idea that this
relation is of dynamical nature. This section is further complemented by a new Figure showing
spatial maps of correlations between the local NST/NSS variability and NST/NSS variability
at Ifaty as inferred from the model. These maps emphasize that co-variability is not only
restricted to the AC core region but occurs for the wider AC region.

**4) Question on lines 246 onwards: Reviewer asked if correlations between model SST and observations were done for the Ifaty coral site, the AC region or the SW Indian Ocean more broadly**

Answer: The answer to this question can be found in lines 230-235 where we have defined the regions used, it reads: "To further validate our hypothesis that the Sr/Ca and $\delta^{18}O_{seawater}$ records from the Ifaty-Tulear reef complex are representative for temperature and salinity in the wider AC region, we analysed the relationship between the temporal evolution of annual mean (January to December, changed in the revised version to "March to February") **salinity and temperature at the location of Ifaty (43°E, 23°S) and within the AC (30°E, 32°S) in a hindcast simulation** with the mesoscale eddy-rich ocean/sea-ice model configuration INALT20 (Schwarzkopf et al., 2019), as well as in SODA and additional reanalysis and observation-based products (EN4 and HadISST; Good et al., 2013; Rayner et al., 2003)."

**5) More detailed description of interannual and decadal variability…explore links with ENSO, PDO etc.**

Answer:
We will explore further the links with ENSO and PDO via EOF and running correlation analysis. Regarding ENSO's influence on the region, it has been shown that only 10-20% of variability in SST or current transport is explained by ENSO (Paris et al., 2018). Our analysis and discussion had, therefore, mainly focused on the suggestion by earlier studies on Agulhas Current and leakage SST and salinity showing a lagged response to ENSO up to 24 month. Those results were based on short instrumental observations. Our results provide a long-term assessment far beyond previous assessments. As such, we focused on the lag to ENSO in $\delta^{18}O_{seawater}$, hence salinity. We could confirm that this lag is also observed with the $\delta^{18}O_{seawater}$ data between 1958 and 1995. Now, we can confirm that this lagged response is also observed with the Nino3.4 record based on ERSST5 back to 1880 (see Figure 2 below; r= -0.37, p=0.01). The lagged response is also reproduced with Nino3.4 paleoclimate reconstructions back to 1750, yet only significant at the 90% level (r= -0.2, p=0.1).
We avoided overinterpretation of ENSO's influence in our discussion mainly focusing on comparison of the interannual frequencies in our record and how that compares to previous studies for the Indo-Pacific Ocean to draw some careful conclusions regarding potential influences of ENSO in the pre-industrial period. Earlier work by Zinke et al. (2004) already concluded that the relationship between coral $\delta^{18}O$ and ENSO was non-stationary. Thus, drawing conclusions on ENSO's influence beyond the instrumental era is difficult. The latter is mainly due to Last Millennium ENSO reconstructions still not agreeing on the sign and variability (Emile-Geay et al., 2013; Steiger et al., 2018). We had, therefore, opted to tone down the discussion of ENSO's role in the region.
Now, we have tested band pass filtering of the Ifaty-Tulear $\delta^{18}O_{seawater}$ and palaeo-ENSO reconstructions for interannual to multi-decadal periodicities as well as running correlations (see Figure 3 below; Emile-Geay et al., 2013; Steiger et al., 2018). The ENSO reconstructions do not agree with each other for large parts of the record since 1661. The best agreement is found for the period where both ENSO reconstructions were calibrated with instrumental data (1870-1995) for the 3.3 to 4 year frequency band. Consequently, our band pass filtered $\delta^{18}O_{seawater}$ record showed various levels of agreement and disagreement with individual ENSO reconstructions. Running correlations (31-year) revealed a highly non-stationary relationship between Ifaty $\delta^{18}O_{seawater}$ and ENSO, switching between negative and positive correlations (see Figure 4 below). Yet, spectral coherence analysis suggests that Ifaty-Tulear $\delta^{18}O_{seawater}$ is coherent with the Nino3.4 index for observations and paleo reconstructions at frequencies between 3.3 and 4 years, as well as decadal bands ranging between 13-30 years. We will include this analysis in the Supplements. Further results on the lagged correlations are discussed below and explored in Figures 5 and 6 of this response. The results show that the 24-month lagged correlation between $\delta^{18}O_{seawater}$ and Nino3.4 is persistent for the majority of the record. Uncertainties in ENSO reconstructions and/or in our $\delta^{18}O_{seawater}$ record may have affected the lagged correlations beyond 1750. Nevertheless, the consistent lagged response to ENSO is most likely the most important finding of this study.

[Figure]

**Figure 3** – Band pass filtered data for Ifaty-Tulear $\delta^{18}O_{seawater}$ (blue), Nino3.4 index from Emile-Geay et al. (2013; EG13; red) and Nino3.4 index from Steiger et al. (2018; black; PHYDA) for a) interannual (3.3 to 3.9 years), b) interdecadal (15-30 years) and c) multidecadal (40-66 years) frequency bands.

[Figure]

**Figure 4** – 31-year running correlations (black line) between Ifaty-Tulear $\delta^{18}O_{seawater}$ composite and a) Nino3.4 reconstructions of Steiger et al. (2018; PHYDA), b) Nino3.4 reconstruction of Emile-Geay et al. (2013; EG13) and c) PDO reconstruction from instrumental ERSST5 data (1880-1995). Grey lines mark 95% confidence interval. Grey shaded bars highlight period of significant negative correlations. Overall, the relationships are highly non-stationary.

The PDO has been suggested to play a small, yet important role in SW Indian Ocean SST and
rainfall on interdecadal time scales (Crueger et al., 2009; Grove et al., 2013).) The same holds
for correlations between the PDO and Ifaty-Tulear ERSST5 (r=0.26, p=0.003). Crüger et al.
(2009) showed that the combined SST and SLP patterns related to Pacific Decadal Variability
has some influence on the Ifaty coral $\delta^{18}$O-SST by influencing trade winds and the South
Equatorial Current. A 31-year running correlation between the PDO index based in ERSST5
and $\delta^{18}$O$_{seawater}$ revealed a non-stationary relationship (see Figure 4c above). The correlation
coefficient for the entire record between 1880 and 1995 is r= -0.28 (p=0.01), thus relatively
weak. Negative correlations ranging between -0.4 and -0.6 were observed for 31-year periods
centered around 1900, 1940 and 1970. In these periods, a negative PDO was associated with
positive $\delta^{18}$O$_{seawater}$ anomalies (more saline conditions). In the revised version we will further
investigate the PDO influence during the instrumental data period to assess if further
conclusions can be drawn with regard to decadal variability observed in our Ifaty $\delta^{18}$O$_{seawater}$
record.
We plan to include the figures in this response as Supplementary Figures in the revised
manuscript.
**6) Question about the comparison of regional $\delta^{18}$O$_{seawater}$ reconstructions across the**
**western Indian Ocean and why correlations are not significant.**
Answer:
We agree with the reviewer that we could have pointed out that while long-term changes agree,
year to year variability does differ between reef sites. We will amend the text accordingly. The
caption of Fig. S9 does show the correlation and p-value for Antongil Bay with Mayotte and
indicates that other correlations between sites are not significant.
We added to caption of Fig. 10: "Only the correlation between Mayotte and Antongil Bay
reconstructed $\delta^{18}$O$_{seawater}$ is statistically significant."
**Technical comments:**
Line 184: Are these GPS coordinates for coral sites?
Answer: No, there are the grid-box GPS coordinates which include our coral sites.
Lines 191-192: Why was the MEI index used?
Answer: The MEI index is a superior index to Nino3.4 because it reflects the combined
atmosphere-ocean (multivariate) signature of ENSO which influences the Indian Ocean.
Furthermore, the MEI index has been used by previous studies which assessed the salinity-
ENSO relationship of the Agulhas Current and leakage region, so for optimal comparison we
opted for the MEI index as well. However, using Niño3.4 index instead of the MEI index leads
to the same conclusion (see figure 3 below for detrended data). The Niño3.4 data show a lead
of 20-24 month to $\delta^{18}$O$_{seawater}$ with a negative correlation in agreement with earlier studies for
the Agulhas leakage region salinity between 1880 and 1995 (see figure 5 below). For the period
1950 to 1995, the 20-24 month lagged correlation is even higher (r= -0.42, p=0.006).
We also tested the relationship with two Nino3.4 paleo-reconstructions of Emile-Geay et al.
(2013) and Steiger et al. (2018) (see Fig. 5b, c and Fig. 6 below). The lagged correlation of 24
month is confirmed back to 1880 in all Nino3.4 indices and back to 1750 with the Steiger et al.
(2018) and Emile-Geay et al. (2013) reconstructions (Fig. 6a, b). However, the lagged correlation is stronger between 1880 and 1995 then pre-1880. We plan to include the figures
as Supplementary Figures in the revised manuscript.

[Figure]

**Figure 5** – Lagged correlation between annual mean Ifaty-Tulear $\delta^{18}O_{seawater}$ composite with
Nino3.4 index between 1880 and 1995 from a) based on ERSST5, B) from Steiger et al. (2018;
PHYDA) and c) from Emile-Geay et al. (2013: EG13). Negative lag means Nino3.4 is leading.
The analysis confirms the lag of 15-24 month between Nino3.4 and Ifaty-Tulear $\delta^{18}O_{seawater}$
beyond 1958.

Jul-Jun averaged Niño34 PHYDA vs Ifaty $\delta^{18}O_{SW}$ composite 1750:1995

[Figure]

Jul-Jun averaged Niño34 EG13 vs Ifaty $\delta^{18}O_{SW}$ composite 1750:1994

[Figure]

**Figure 6** - Lagged correlation between annual mean Ifaty-Tulear $\delta^{18}O_{seawater}$ composite with Nino3.4 index between 1750 and 1995 from a) Steiger et al. (2018; PHYDA) and b) Emile-Geay et al. (2013; EG13). For this period, correlations are significant at 90% level or higher. Beyond 1750, the 15-24-month lagged correlation is no longer significant.

**Other comments:**

Lines 202-203: Difficult to see how well ERSST5 compares to Sr/Ca-SST in Fig. 2.

Answer: We have illustrated the 1850-1995 record in Fig. S1 in the Supplements to enable a direct visual comparison for the instrumental era. Figure 2 in the paper is to show the long-term coral record with the instrumental data overlaid (*e.g.* ERSST, SODA salinity). We refer the reader to Fig. S1 to get a better idea of the match and mismatch periods for the instrumental era. In addition, Tab. S1 shows all correlation coefficients and significance levels. As stated earlier, we cannot expect a close agreement between ERSST5 and Sr/Ca-SST for all data periods pre-1970 due to very sparse observations.

Line 207: What are the slopes for the calibration equations used in Tab. S1?
Answer: As specified in the methods section, we did not perform a calibration. We applied the
mean slope for Sr/Ca vs. SST of -0.06mmol/mol/°C following Correge (2006) and Pfeiffer et
al. (2017) on Sr/Ca and $\delta^{18}$O anomalies (relative to 1961 to 1990) and we randomly propagated
the slope errors (±0.01mmol/mol °C$^{-1}$) based on literature estimates in our Monte Carlo
reconstruction. It is specified in the methods section.
Lines 215-217: Equations for $\delta^{18}$O$_{seawater}$ reconstruction should be presented…
Answer: The equations are now included in the methods section 2.3.
Lines 228-230: "I'm having a hard time understanding the sentence…"
Answer: Here we mean that the interannual and decadal variations between fresher and saltier
periods pre-1970 indicated by $\delta^{18}$O$_{seawater}$ are mostly positively correlated (not significant) with
Sr/Ca-SST and instrumental SST. Yet, statistically no clear causal relationship could be
established. We have clarified this in the text. It now reads: "For the record between 1854 and
1995, it appears as if decreasing (increasing) Ifaty-Tulear $\delta^{18}$O$_{seawater}$, i.e., freshening
(salinification), coincides with decreasing (increasing) Sr/Ca-SST and ERSST5, i.e., cooling
(warming). Yet, the relationship is weak and interannual to decadal variability is not
statistically significant correlated. Hence, no robust correlation or causality could be
established between the temporal evolution of regional temperature and salinity."
Lines 225: "What's the correlation and significance between SODA salinity and the $\delta^{18}$O$_{seawater}$
record?
Answer: Table 1 shows all correlations and significance levels, and the 95% confidence
intervals for these correlations. The correlation is 0.63 (0.50 for detrended data) with
significance ranging between p=0.008 and 0.001. The correlations with AC region salinity are
higher at r= 0.7, p<0.001 (r= 0.57 and p=0.002 for detrended data).
Lines 264-267: explain what positive correlation between d18Osw and rainfall means etc.
Answer: Now added: "Rainfall and salinity or $\delta^{18}$O$_{seawater}$ should be negatively correlated when
rainfall or freshwater runoff influences the signal."
Lines 282-282: …what a positive correlation means in terms of how changes in zonal wind
stress impact $\delta^{18}$O$_{seawater}$ …
Answer: It now reads: "Our low-pass filtered reconstructed $\delta^{18}$O$_{seawater}$ record indicates a
positive correlation (r= 0.67, p= 0.0063) with the southern Indian Ocean (10-40°S, 50-100°E)
ICOADS zonal wind stress, pointing to easterly wind anomalies driving ocean advection of the
salinity signal across southern Indian Ocean."
Lines 285-287: It would be helpful to indicate the region you are referring to in Fig. S7

Answer: That region is already specified in the text. The regions we mean are highlighted in colour, they represent the regions with significant correlations. We have now added rectangles for the regions we refer to.

Lines 313-315: "I don't agree that Fig. S8 demonstrates that $\delta^{18}O_{seawater}$" covaries with regional rainfall.

Answer: With regional rainfall we mean the rainfall around the coral site or nearby land regions. We do not imply a large-scale correlation here. We now clarified that in the text.

Figure 2: Make dark red line darker

Answer: Done.

Figure 4: Caption difficult to understand

Answer: We reformulated the caption and hope it reads better now: "**Reconstructed and**

**simulated co-variability of temperature and salinity in the Ifaty-Tulear and AC core**

**regions**. (a-c) SST at Ifaty reconstructed from coral Sr/Ca (red), simulated with INALT20

(black), and obtained from ERSST5 (dark yellow), as well as SST in AC core region simulated with INALT20 (grey) and obtained from ERSST5 (light yellow); (d-f) SSS at Ifaty reconstructed from coral d18Osw (blue), simulated with INALT20 (black), and obtained from

SODA (dark cyan), as well as SSS in AC core region simulated with INALT20 (grey), and obtained from SODA (light cyan). Shown are annual mean (thin lines) and sub-decadally filtered (7-year Hamming filter) anomalies (referenced to 1961-1990 mean), whereby annual means in ocean model and instrumental data are calculated as March to February averages for better comparison with the coral record."

Figure 5: Please use colour.

Answer: We increased the contrast of black and grey lines in Figure 5. It is consistent with

Figure 4.

Figure 7: Use same Y-axis on all panels, explain green lines.

Answer: Done. Green lines changed to grey and mean correlation to black line. Grey line legend now included stating that it shows the 95% confidence interval of the correlation.

Figure S1: add a, b, c and d

Answer: Done.

Figure S2: It seems that annual $\delta^{18}O_{seawater}$ record underestimates seasonal extremes….

Answer: We dont have high-resolution data here, so we need to rely on the annual record at hand. This figure is to illustrate that the annual record captured the year to year variability for the majority of the bimonthly record data. We do not attempt to capture all seasonal extremes by annual data, rather focus on the interannual to decadal changes and agreements on long- term trends.

Figure S3: Please use colour in panel b

Answer: Done.

Figure S8: Can't read any of the small text at top of panels.

Answer: Done.

---

## Author Comment (AC2)

**Point-by-point response**

We thank both reviewers for their constructive comments which we address briefly in this response.

**Reviewer 2:**

**1) Add details on which samples were drilled in previous studies and which ones for this study…some fine tuning of wording around use of growth banding or high-resolution oxygen isotopes profiles**

Answer:

For information, current methods states: "We resampled the Ifaty-4 core at annual resolution for Sr/Ca, except for multidecadal periods subsampled previously at bimonthly resolution (Zinke et al., 2004) following the established and precise age model of the high-resolution $\delta^{18}O$ sampling from austral summer to summer in any given annual cycle. Cores Ifaty-1 and Tular-3 were sampled at annual resolution along the major growth axis following the density pattern from summer to summer in any given annual cycle, established from X-ray-radiograph-positive prints. "

We will clarify the use of previous and new data in a Supplementary Table.

**2) Simplify description of Monte Carlo approach for seawater oxygen isotope reconstructions…clarify if Monte Carlo approach was also used for 1881-1661 section.**

Answer:

See our comment to Rev. 1.

The Monte Carlo approach was used for all data including the 1881 to 1661 section, as described in methods.

**3) Why was average Sr/Ca-SST slope used? Why HadISST for $\delta^{18}O_{seawater}$ reconstruction and not ERSST?**

Answer:

We did not use the average proxy-SST slopes alone, the Monte Carlo approach applies slope errors randomly. It reads: "Monte Carlo parameters are calculated by adding random values on the proxy-SST slopes, Sr/Ca, and $\delta^{18}O$ (random values are normally distributed numbers in the 1 σ range of slope errors and analytical errors, respectively)." Thus, we fully take into account uncertainties in slope estimates reported in the literature.

HadISST was used to cross-check the $\delta^{18}O_{seawater}$ reconstruction based on Sr/Ca-SST with a different SST dataset then ERSST5. By doing so, we assessed if results would improve by using reconstructed SST from observations at 1x1 degree spatial resolution instead of Sr/Ca-SST for a longer period of time. We have done the $\delta^{18}O_{seawater}$ reconstruction with ERSST5 as well, see Figure 2 of this response. We observe close agreement in long-term changes for the majority of the past 140 years (see Figure 2 of this response).

**4) Improve discussion of model results in comparison to coral-based reconstructions.**

Answer: See response to reviewer 1 above.

**Minor comments:**

Line 17: Might be helpful to define the acronyms for sea-surface temperature and salinity as they're used later in the abstract.

Answer: Done.

Line 22: please indicate the full time period of comparison (1958-1995?)

Answer: Done.

Line 38: both "inter-ocean" and "interocean" appear in the manuscript. Use one or the other for consistency.

Answer: inter-ocean now used consistently

Line 42: possible formatting issue on one of the references?

Answer: Corrected.

Line 78: This is the first mention of $\delta^{18}O$. It might be "spelling out" what the $\delta^{18}O$ notation stands for.

Answer: Changed to "…Measurements of the $\delta^{18}O$ in seawater (hereafter $\delta^{18}O_{seawater}$),…"

Line 170: One occurrence of "for SST" can be removed.

Answer: Corrected.

Line 160: The -0.22 per mil/deg C relationship pre-dates Thompson et al., 2011. Please use the appropriate reference here.

Answer: Added: Lough, 2004.

Lines 202-205: Interestingly the $\delta^{18}O$-SST variability appears to be more consistent with ERSST than Sr/Ca-SST (which has some very large spikes that aren't observed in temperature). Any thoughts on why this is the case?

Answer:
Up to 1890, $\delta^{18}O$-SST apparently agrees better with ERSST than Sr/Ca-SST. Pre-1890, $\delta^{18}O$-SST deviates from ERSST more than Sr/Ca-SST. Between 1942 to 1995, both proxies perform equally well. Between 1854 and 1910, Sr/Ca-SST outperforms $\delta^{18}O$-SST, most probably due to greater impacts of $\delta^{18}O_{seawater}$ variations (already suggested by Zinke et al., 2004). Especially between 1910 and 1940, Sr/Ca-SST shows higher magnitude variability for the most recent period. We suggest that Sr/Ca-SST may better reflect local SST variations at the reef site and between reefs which at times may be higher than recorded by the dual proxy $\delta^{18}O$ (influenced by SST and d18Osw) or the coarse gridded ERSST.
As stated in the manuscript, we also expect the annual mean $\delta^{18}O$-SST record to perform better for parts of the 20$^{th}$ century and pre-1890 because 1 core (Ifaty-4) has been previously sampled and measured at higher resolution (monthly 1920-1995; bimonthly 1919-1661) while Sr/Ca is

largely based on annual mean samples for all cores. Thus, annual sampling leads to overall larger uncertainties in reconstructed Sr/Ca-SST for individual years than higher resolution sampling. These uncertainties are propagated by our Monte Carlo approach.

Line 205: Are these trends? If so, please include the term "trend" in the sentence. Also, both numbers are consistent which is nice!

Answer: "trend" now included in sentence

Figure 1: I'd recommend using a different light color to represent the errors in panels a-c (maybe gray) so that the median of each reconstruction is more visible. This is more of an issue with the panel A where the shades of red are very close to each other.

Answer: We have changed the median line in panel a. See comment to Rev. 1.

Line 240+: the use of both NST and SST is confusing. Using NST alone for this presentation is fine. Same goes for NSS/SSS.

Answer: NST is the correct description of model data. We have now clarified in methods why we use NST instead of SST for the model data.

Line 330: This sentence might be missing a few words?

Answer: Unclear what the reviewer refers to.

Line 395: This sentence might be too strong and casts a lot of doubt the observations in the rest of the paragraph, especially given the evidence from the literature presented in the next paragraph that supports more ENSO activity in the 16th century.

Answer: We have referred to studies that show enhanced interannual ENSO-band variability in the 17th century and at the turn to the 18th century, not the 16th century. It's unclear what the reviewer refers to here.

Line 400: Cobb 2003 is a more appropriate reference

Answer: Cobb, 2003 added

Line 402: Is this the same coral used in this study? If so, using it to support the results is somewhat circular. If it's a different core, it might be worth mentioning so others don't make the same assumption.

Answer: We now make it clear that it is related to the previous study.

---

## Author Response (AR1)

**Point-by-point response**

We thank both reviewers and the Editor for their constructive comments which we address in this response.

In our initial response to both reviewers, we had already answered most questions. We therefore refer to the document "Response to Reviewer 1 and 2" uploaded online in addition to this response. Here we address some issues in further detail.

Reviewer 1

**1) …why we used annual resolution Sr/Ca record and how to explain relatively low correlation with SST…**

In our initial response to the reviewer, we had already outlined this answer. We therefore refer to the document "Response to Reviewer 1" uploaded online.

We have provided additional support for the validity of our annual coral Sr/Ca-SST and its correlation with ERSST5 and other SST products. Watanabe et al. (MS in prep.) have adopted the approach of Smerdon et al. (2016) testing the signal to noise ratio in proxy and instrumental SST (**see below Fig. 1**). Our correlation ranges between 0.3 and 0.4 in line with the expected correlation given the regional signal noise ratio. This is now included in the section 4.1. of the discussion, which reads:

"Furthermore, the standard deviations of mean annual SST at Ifaty is only 0.25°C which leads to a lower signal to noise ratio in annual Sr/Ca-SST estimates. With Sr/Ca-SST having an analytical uncertainty of ±0.15°C, the correlation between ERSST and coral Sr/Ca-SST should range between 0.3 and 0.4 following the method of Smerdon et al. (2016), exactly what we obtained in this study."

Furthermore, we have included a new Figure S3 that shows the number of SST observations drastically decling beyond 1970.

**2) Agreement/disagreement between $\delta^{18}O_{seawater}$ based on Sr/Ca-SST vs. using HadISST…**

This question was also answered in our initial response to reviewer 1. We have amended Figure 3a by showing the reconstructed $\delta^{18}O_{seawater}$ based on ERSSTv5 instead of HadISST as suggested by the reviewer. Figure S5 shows both $\delta^{18}O_{seawater}$ based on ERSSTv5 and HadISST in comparison to $\delta^{18}O_{seawater}$ based on Sr/Ca-SST.

**3) Manuscript could benefit form more detailed description of model results and potentially re-framing of the aims for the model study…**

We have modified sections 3.1 and 3.2 as suggested in our response to reviewer 1. The new section 3.1 "Validation of reconstructed Sr/Ca-SST and d18Oseawater at Ifaty" is focusing on a comparison of the coral reconstructions for SST and SSS variability at Ifaty with available gridded observation-based products and model data in the Ifaty-Tulear region. It includes a

discussion of the discrepancy between the different products regarding the exact temporal evolution of SST and SSS caused by limited number of observations and highlights the best agreement of the coral data with ERSST and SODA. The new section 3.2 "Representativeness of SST and SSS variability at Ifaty for variability in the wider Agulhas Current region" then focuses on potential co-variability between SST and SSS at Ifaty and other locations in the wider Agulhas region. Here, independent of the mentioned disagreements in the exact temporal evolution, all observational products as well as the model agree that variability at Ifaty is indeed representative for variability in the AC core region (Figure 3). The fact that co-variability is not only found in observation-based products but also in the simulated NST and NSS from an ocean model without data assimilation, supports the idea that this relation is of dynamical nature. This section is further complemented by a new Figure 3b showing spatial maps of correlations between the $\delta^{18}O_{seawater}$ based on Sr/Ca-SST and SODA salinity. This map emphasize that co-variability is not only restricted to the AC core region but occurs for the wider AC region.

[Figure]

Figure 1 – Monte-Carlo (left) and bootstrap method (right) to estimate the distribution of correlation coefficient. Upper panels coral Sr/Ca-SST compared to ERSSTv5 and lower panels same but for HadISST. The red dot lines show 95 percentiles. As the lower border of 95 percentiles is higher than zero, the correlation coefficient is significant at a 95% confidence level (Watanabe et al., MS in prep.).

**4) Question on lines 246 onwards: Reviewer asked if correlations between model SST and observations were done for the Ifaty coral site, the AC region or the SW Indian Ocean more broadly**

Question was answered in our initial response.

**5) More detailed description of interannual and decadal variability…explore links with ENSO, PDO etc.**

We have expanded the analysis of ENSO teleconnections with Ifaty $\delta^{18}O_{seawater}$ based on Sr/Ca-SST in Figure 7g and h, Figure 9 b to d and a new Figure 10. These results are now discussed in the revised manuscript. Figures S10, S12 and S13 complement this expanded analysis.
Our wavelet coherence analysis shown in new Figure 9 indicates coherence with ENSO reconstructions on interannual and interdecadal frequencies, the latter involving phase lags. The 2-4 years and 8-16 years frequency bands appear to be important time scales of ocean climate variability in the greater Agulhas region.
We aim to avoid overinterpretation of ENSO's influence in our discussion mainly focusing on comparison of the interannual and decadal frequencies in our record and how that compares to proxy-based ENSO reconstructions. We draw some very careful conclusions regarding potential influences of ENSO in the pre-industrial period yet like to state here that ENSO alone may not be the only driver influencing salinity, $\delta^{18}O_{seawater}$ or SST. The results show that the 24-month lagged correlation between $\delta^{18}O_{seawater}$ and Nino3.4 is persistent for the majority of the record. Uncertainties in proxy-based ENSO reconstructions beyond 1880 and/or in our $\delta^{18}O_{seawater}$ record may have affected the lagged correlations beyond 1750. Nevertheless, the consistent lagged response of salinity and $\delta^{18}O_{seawater}$ to ENSO is most likely the most important finding of this study.

**6) Question about the comparison of regional $\delta^{18}O_{seawater}$ reconstructions across the western Indian Ocean and why correlations are not significant.**

We have expanded our analysis and interpretation of the comparison of regional $\delta^{18}O_{seawater}$ reconstructions across the western Indian Ocean. A new Figure S15 (next to Figure S14) shows the actual differences between individual $\delta^{18}O_{seawater}$ records to Ifaty $\delta^{18}O_{seawater}$. This analysis reveals that the absolute difference is smaller than the individual uncertainties from the reconstruction method, thus the $\delta^{18}O_{seawater}$ ranges for all western Indian Ocean sites fully overlap and are indistinguishable.

**Technical comments:**

Here we address the changes we made that were not yet addressed in our initial response.

Lines 191-192: Why was the MEI index used?

We now also show the same lagged correlation with the instrumental Nino3.4 index (back to 1880) and the Nino3.4 paleo-reconstructions of Emile-Geay et al. (2013) and Steiger et al. (2018) in our modified Figure 7 and Figures S10 and S12.

Lines 215-217: Equations for $\delta^{18}O_{seawater}$ reconstruction should be presented…

Done. Section 2.4 has been modified.

Lines 264-267: explain what positive correlation between d18Osw and rainfall means etc.
Done. Now added: "Rainfall and salinity or $\delta^{18}O_{seawater}$ should be negatively correlated when rainfall or freshwater runoff influences the signal."

Lines 282-282: …what a positive correlation means in terms of how changes in zonal wind stress impact $\delta^{18}O_{seawater}$ …
Done. It now reads: "Our low-pass filtered reconstructed $\delta^{18}O_{seawater}$ record indicates a positive correlation (r= 0.67, p= 0.0063) with the southern Indian Ocean (10-40°S, 50-100°E) ICOADS zonal wind stress, pointing to easterly wind anomalies driving ocean advection of the salinity signal across southern Indian Ocean."

Figure 2: Make dark red line darker
Answer: Done.
Figure 4: Caption difficult to understand
Done. We modified the caption.

Figure 5: Please use colour.
We increased the contrast of black and grey lines in Figure 5. It is consistent with Figure 4.

Figure 7: Use same Y-axis on all panels, explain green lines.
Figure 7 has been modified and expanded. Red and green lines are replaced by black and grey lines for clarity.

Figure S1: add a, b, c and d
Answer: Done.

Figure S8: Can't read any of the small text at top of panels.
Answer: Done.

**Reviewer 2:**

**1) Add details on which samples were drilled in previous studies and which ones for this study…some fine tuning of wording around use of growth banding or high-resolution oxygen isotopes profiles**

We added the data in a new Supplementary Table S1.

**2) Simplify description of Monte Carlo approach for seawater oxygen isotope reconstructions…clarify if Monte Carlo approach was also used for 1881-1661 section.**

Done. Entire section was modified.

**3) Why was average Sr/Ca-SST slope used? Why HadISST for $\delta^{18}O_{seawater}$ reconstruction and not ERSST?**

We now shows $\delta^{18}O_{seawater}$ based on ERSSTv5 In Figure 3a and Figure S5.
For the slope, please refer to our initial response to reviewers comments.

**4) Improve discussion of model results in comparison to coral-based reconstructions.**

Done. See response to Reviewer 1 above.

**Minor comments:**

Line 17: Might be helpful to define the acronyms for sea-surface temperature and salinity as they're used later in the abstract.

Answer: Done.

Line 22: please indicate the full time period of comparison (1958-1995?)

Answer: Done.

Line 42: possible formatting issue on one of the references?

Now corrected.

Lines 202-205: Interestingly the $\delta^{18}O$-SST variability appears to be more consistent with ERSST than Sr/Ca-SST (which has some very large spikes that aren't observed in temperature). Any thoughts on why this is the case?

Please see our response to reviewer 1 above. And please have a look at the estimation of correlation we present in this letter which takes into account the signal to noise ratio between ERSSTv5 and coral Sr/Ca-SST following Smerdon et al. (2016). We also expanded the discussion.

Line 400: Cobb 2003 is a more appropriate reference

Cobb, 2003 now added.

1. Your response clarifies reviewer #2s question about methods for sampling annual average coral material, though it would be good to understand more about the potential uncertainties in this approach. For example what is the uncertainty on the timing of the annual d18O and annual density markers (and I would expect that the uncertainty is larger for density-based approaches), and could you run some tests to see what these levels of uncertainty do in terms of reducing correlations between true annual averages and coral-cycle derived pseudo-annual averages?

We have computed the pseudocoral annual averages and performed a Monte Carlo simulation to test for age model uncertaintties in annual mean sampling based on density banding. We introduced a new section "2.3. Age model uncertainty" where we outline the approach. New Figures S1 and S2 illustrate the results.
It reads: "The difference of Sr/Ca between the true and pseudo values is 0.003 ± 0.007 mmol/mol (1σ) (i.e., about ± 0.1 ºC) while the difference in $\delta^{18}O$ is 0.02 ± 0.014‰. Because of SST-related seasonality, Sr/Ca and $\delta^{18}O$ may have a bias towards positive values (lower SST), yet this bias is low. $\delta^{18}O_{sw}$ estimated from paired coral $\delta^{18}O$ and Sr/Ca measurements (see section 2.4 for methodology) is not significantly affected by the age model error (0.00±0.03‰ between true and pseudo values)."

2. The reviewers are both keen to see more description/analysis of the processes by which ENSO variability manifests at your site. It would be good to elaborate on this in the text and possibly with an additional figure.

Please see our response to the reviewers in the point-by-point response above. We have modified the figures 7 and 9 and included a new Figure 10 to show the results of band-pass filtering, cross-spectral analysis and wavelet coherence analysis.

3. One of the really important contributions of this work is a multi-century d18O-sw reconstruction, and it would be good to include some additional interpretation of the long-term trends/changes in this. In addition to the comparisons with similar reconstructions from other Indian Ocean sites it would be good to also show a comparison of your record with other long-term reconstructions of d18O-sw, including from other ocean basins. This could give some interesting perspectives to long-term drivers of change.

We have tested the relationship between Ifaty $\delta^{18}O_{seawater}$ and $\delta^{18}O_{seawater}$ reconstructions from the central Pacific Line Islands. There is no correlation between the records. In our opinion this comes as no surprise. It is already complicated to draw conclusions about relationships between Indian Ocean $\delta^{18}O_{seawater}$ records. Our analysis of all published western Indian Ocean

$\delta^{18}O_{seawater}$ records (Fig. 11; Figs. S14 and S15) demonstrates that the absolute difference is smaller than the individual uncertainties from the reconstruction method, thus the $\delta^{18}O_{seawater}$ ranges for all western Indian Ocean sites fully overlap and are indistinguishable. We propose that $\delta^{18}O_{seawater}$ is modified by site-specific atmospheric (P-E) and oceanic variability, and likely involve temporal lags.

We have, however, now included the comparison to two established Palaeo-Nino3.4 reconstructions and show the relationship over the full record in our modified Figures 7, 9, 10, S10 and S12.

---

## Author Response (AR3)

**Point-by-point response**

We thank both reviewers and the Editor for their constructive comments. Below we address the final points raised by the editor.

*Obtain a doi for the data repository page where the data associated with this study will be archived, and add this to the data availability section of your paper

Answer: We have submitted the data to the NOAA database, yet did not receive a doi yet.

*Remove the two test references from the start of the reference list

Answer: Done.

*Have a thorough review of figures, ensuring consistency in style and legible labels considering the sizing of figures in the final publication.

Answer: We have checked all figures to have consistent font styles and labels.

*I also feel that there are too many figures, many with multiple panels so that it is difficult for readers to get a clear understanding of your main findings. Please consider which figures are necessary for the main text, and which are better placed in the supplementary material (I would suggest figures 5, 7, 8, 9 might be better placed in the supplement). There may also be supplementary figures that are superfluous, but i will leave it for you to decide if all are necessary

Answer: We have moved Figures 5 (now Fig. S8) and 8 (now Fig. S11) into Supplements to reduce the number of figures in the main manuscript file. We have not changed the other Supplementary Figures as we see them as important.

*Please invert the y-axis on panel c of figure 2, so that there is consistency across all of your figures in the up-down direction for d18osw/SSS.

Answer: Done.